# Synchronous multi-segmental activity between metachronal waves controls locomotion speed in *Drosophila* larvae

Yingtao Liu[1,2], Eri Hasegawa[2†], Akinao Nose[1,2], Maarten F Zwart[3*], Hiroshi Kohsaka[2,4*]

[1]Department of Physics, Graduate School of Science, The University of Tokyo, Tokyo, Japan; [2]Department of Complexity Science and Engineering, Graduate School of Frontier Science, The University of Tokyo, Kashiwa, Japan; [3]School of Psychology and Neuroscience, Centre of Biophotonics, University of St Andrews, St Andrews, United Kingdom; [4]Graduate School of Informatics and Engineering, The University of Electro-Communications, Tokyo, Japan

**Abstract** The ability to adjust the speed of locomotion is essential for survival. In limbed animals, the frequency of locomotion is modulated primarily by changing the duration of the stance phase. The underlying neural mechanisms of this selective modulation remain an open question. Here, we report a neural circuit controlling a similarly selective adjustment of locomotion frequency in *Drosophila* larvae. *Drosophila* larvae crawl using peristaltic waves of muscle contractions. We find that larvae adjust the frequency of locomotion mostly by varying the time between consecutive contraction waves, reminiscent of limbed locomotion. A specific set of muscles, the lateral transverse (LT) muscles, co-contract in all segments during this phase, the duration of which sets the duration of the interwave phase. We identify two types of GABAergic interneurons in the LT neural network, premotor neuron A26f and its presynaptic partner A31c, which exhibit segmentally synchronized activity and control locomotor frequency by setting the amplitude and duration of LT muscle contractions. Altogether, our results reveal an inhibitory central circuit that sets the frequency of locomotion by controlling the duration of the period in between peristaltic waves. Further analysis of the descending inputs onto this circuit will help understand the higher control of this selective modulation.

***For correspondence:**
mfz@st-andrews.ac.uk (MFZ);
kohsaka@edu.k.u-tokyo.ac.jp
(HK)

**Present address:** [†]Frontier Science and Social Co-creation Initiative, Kanazawa University, Kakuma, Kanazawa, Ishikawa, Japan

**Competing interest:** The authors declare that no competing interests exist.

## Editor's evaluation

Exploiting the power of the *Drosophila* larva as a model, this fundamental study sheds light on the neuronal mechanisms of speed regulation during locomotion. The data obtained using a combination of functional and structural approaches are rigorous and convincing. The identified mechanism of speed regulation could be shared with limbed animals and therefore this work is of relevance to those studying the motor system and locomotion across species.

## Introduction

Animals flexibly adapt their speed of locomotion to meet their behavioral needs (*Alexander, 1989*; *Byrne, 2019*; *DeAngelis et al., 2019*). In recent decades, the neural basis of the modulation of the speed of locomotion across the animal kingdom has received much attention. The mesencephalic locomotor region (MLR), which projects to reticulospinal neurons that in turn innervate spinal circuits, has been identified in all vertebrate species studied to date as an important control center (*Ryczko*

*et al., 2017*). Increasingly intense stimulation of the MLR causes increases in the speed of locomotion, with accompanying gait transitions (*Atsuta et al., 1990*; *Grillner, 1985*; *Shik et al., 1966*; *Shik and Orlovsky, 1976*; *Skinner and Garcia-Rill, 1984*). The spinal cord recruits different types of motor neurons at different speeds, with the accompanying changes in gait requiring widespread reconfiguration within its circuitry (*Dasen, 2017*; *Kiehn, 2016*).

How does the central nervous system (CNS) vary the frequency of locomotion to achieve the required speeds? In a range of species, descending projecting excitatory neurons have been shown to drive the rhythm of locomotion (*Berg et al., 2018*; *Caggiano et al., 2018*; *Capelli et al., 2017*; *Friesen and Kristan, 2007*; *Gatto and Goulding, 2018*; *Josset et al., 2018*; *Roberts et al., 2010*). In mice, studies using optogenetics have shown that excitatory neurons are necessary and sufficient for rhythm generation (*Hägglund et al., 2010*; *Hägglund et al., 2013*), with studies ongoing to uncover the precise identity of the rhythm generators (*Kiehn, 2016*). Zebrafish, which use axial locomotion to move, have different central modules corresponding in adults to fast, intermediate, and slow locomotion that are selectively recruited to command the motor pools specific for different speeds (*Ampatzis et al., 2013*; *Ampatzis et al., 2014*). The pacemaker neurons driving locomotion at different speeds have bursting frequencies related to their module affiliation (*Song et al., 2020*).

The kinematics of movement change as a function of frequency depending on the species and gait. Swimming animals modulate their undulatory frequency by controlling the intersegmental lag, which is linearly scaled with the locomotor cycle duration (*Grillner, 1974*). On the other hand, limbed animals change the frequency of walking by varying the locomotor cycle differentially: the stance phase is varied, but the swing phase is almost unchanged, even as animals switch to different gaits. This holds true for animals ranging from insects and tardigrades to mammals (*Boije and Kullander, 2018*; *Frigon et al., 2014*; *Grillner et al., 1979*; *Jacobson and Hollyday, 1982*; *Nirody et al., 2021*). How the nervous system generates this asymmetry in the variation of stance and swing phases is still an open question (*Bidaye et al., 2018*; *Boije and Kullander, 2018*; *Kiehn, 2016*).

Here, we investigated the speed-dependent modulation of locomotion in *Drosophila* larvae and the underlying neural mechanisms. The *Drosophila* larva moves by peristaltic waves, in which body wall muscles contract sequentially from one end to the other (*Berrigan and Pepin, 1995*; *Heckscher et al., 2012*; *Sun et al., 2022*). We found that the *Drosophila* larval locomotor cycle is also differentially modulated: the phase in between each consecutive peristaltic wave (the 'interwave' phase), not the peristaltic wave itself, is primarily varied with speed, reminiscent of the stance phase in limbed locomotion. We then examined the underlying muscular dynamics and found that the interwave phase is characterized by synchronous contractions of the lateral transverse (LT) muscles along the anterior–posterior axis. The amplitude and duration of their contraction scale with the duration of the interwave phase. Using EM connectomics and calcium imaging, we identified two types of interneurons that are associated with the LT neural circuitry and show segmentally synchronized activity: GABAergic premotor neuron A26f and its presynaptic partner GABAergic interneuron A31c. Using optogenetics, we revealed that both A31c and A26f neurons are sufficient and necessary for the desired contraction of the LT muscles and set the speed of locomotion through the modulation of the interwave phase. Our results reveal that the *Drosophila* larva uses a similar strategy to regulate speed as limbed animals by varying the two main phases of the cycle differentially and that the activity of an inhibitory circuit helps to generate this variation.

## Results

### Variability in the interwave phase of crawling contributes to speed variability

Crawling behavior in *Drosophila* larvae is generated by repetitive waves of propagation along the length of their body (*Berrigan and Pepin, 1995*). A previous study in mildly physically restrained first-instar larvae showed that crawling speed correlates with stride period more than stride length (*Heckscher et al., 2012*). It has been shown that the lag between the contraction of adjacent segments during the peristaltic wave (intersegmental lag) scales with the cycle period in the intact animal and the isolated CNS (*Heckscher et al., 2012*; *Lemon et al., 2015*). These observations suggest that the cycle period varies through a uniform, rather than an asymmetric, modulation of the phases of the locomotor cycle. However, physically restricted larvae and the isolated CNS have long cycle periods

(2–20 s), presumably due to aberrant or absent input from sensory neurons (*Caldwell et al., 2003*; *Hughes and Thomas, 2007*; *Schützler et al., 2019*; *Zarin et al., 2019*). How the larva varies locomotion in crawling within the normal range of cycle periods (0.6–2 s) is therefore not understood. We therefore first aimed to address this question (*Figure 1A–I* and *Figure 1—figure supplement 1A–F*). We recorded third-instar larvae freely crawling on an agarose plate and measured the displacement of their body wall segments (*Figure 1A–C*). Larvae crawled at varying speeds even within the same environmental conditions such as temperature (*Figure 1D and E*, 0.35–1.23 mm/s, n = 18 larvae). We found that the locomotion speed in these freely crawling animals also correlated with stride frequency more so than stride length, consistent with a previous report (*Heckscher et al., 2012*; *Figure 1E*). To further characterize the underlying kinematic changes, we assessed how the two previously identified phases within the locomotor cycle (*Heckscher et al., 2012*) change with speed. In the first phase, local body wall contractions are propagated from the posterior to anterior segments (here called 'wave phase'), whereas the second is characterized by the period from mouth parts unhooking to the onset of the tail contraction ('interwave phase'; *Figure 1F*).

Next, we analyzed how crawling speed related to the two locomotor cycle phases (*Figure 1G–I*). We found that both the interwave phase and the wave phase are correlated with crawling speed (*Figure 1—figure supplement 1C–C'*; Pearson correlation coefficient: the wave phase vs. speed $r$ = –0.62, the interwave phase vs. speed $r$ = –0.74). The interwave duration at faster speeds is close to 0; indeed, the duty factor of the interwave phase, which is given by the ratio of the interwave phase duration to stride duration, decreased with speed (*Figure 1I*; linear regression coefficient = –0.47, $r^2$ = 0.55) and was reduced to zero at the faster speeds. The wave and interwave phases are modulated independently, as can be seen by the lack in correlation between these two phases (*Figure 1—figure supplement 1D and E*). These results suggest that the speed-dependent modulation of crawling frequency is largely due to modulation of the interwave phase.

We quantified the duration of each phase in freely crawling larvae. To evaluate the contribution of these two phases to stride duration, we plotted the duration of these phases as a function of stride duration (*Figure 1—figure supplement 1F and F'*). Both are correlated with stride duration, with the interwave phase correlated more strongly than the wave phase (interwave phase $r$ = 0.86, wave phase $r$ = 0.56, p<0.0001). What becomes clear from this analysis is that as the stride duration decreases, the interwave phase shortens while the wave duration stays more or less constant; when the interwave duration becomes minimal, further decreases in stride duration are accompanied by decreases in wave duration. These observations suggest that the interwave phase between peristaltic waves is more variable than wave phase, and that there is a range-dependent modulation of the frequency of locomotion.

## Synchronous contraction of transverse muscles during the interwave phase

To reveal the nature of the interwave phase, we examined the movement of body wall muscles during free crawling. The ends of individual muscles were labeled by expressing GFP in the tendon cells using *sr-Gal4* (*Schnorrer et al., 2007*) and imaged from the side (*Figure 1J and J'*). This allowed us to analyze the contraction dynamics of each muscle in freely crawling larvae. The dynamics of two longitudinal muscles (DO1 and VL4) that span the anterior and posterior boundary of each segment and one transverse muscle (LT2) that runs perpendicular to the anterior–posterior axis of the animal were examined (*Figure 1K*, *Figure 1—figure supplement 1G*, and *Figure 1—videos 1 and 2*). We made three observations: first, consistent with a previous study (*Heckscher et al., 2012*), longitudinal muscles exhibited propagation from the posterior segment to the anterior in forward crawling (*Figure 1L* and *Figure 1—figure supplement 1G'*). Second, the LT2 transverse muscles only showed synchronous contractions across abdominal segments A2-A7 (*Figure 1K* and *Figure 1—figure supplement 1G'*), in contrast to the earlier report (*Heckscher et al., 2012*). Furthermore, while most longitudinal muscles were contracting during peristaltic waves, the transverse muscles contracted exclusively during the interwave phase (*Figure 1—figure supplement 1G'*). Third, the longitudinal muscles in the posterior-most segments (VL4 and DO1 in A6 and A7 in *Figure 1—figure supplement 1G'*) contract during the interwave phase (see 'Discussion'). Taken together, longitudinal muscles predominantly show sequential activity during the wave phase, whereas transverse muscles exhibit synchronous activity during the interwave phase.

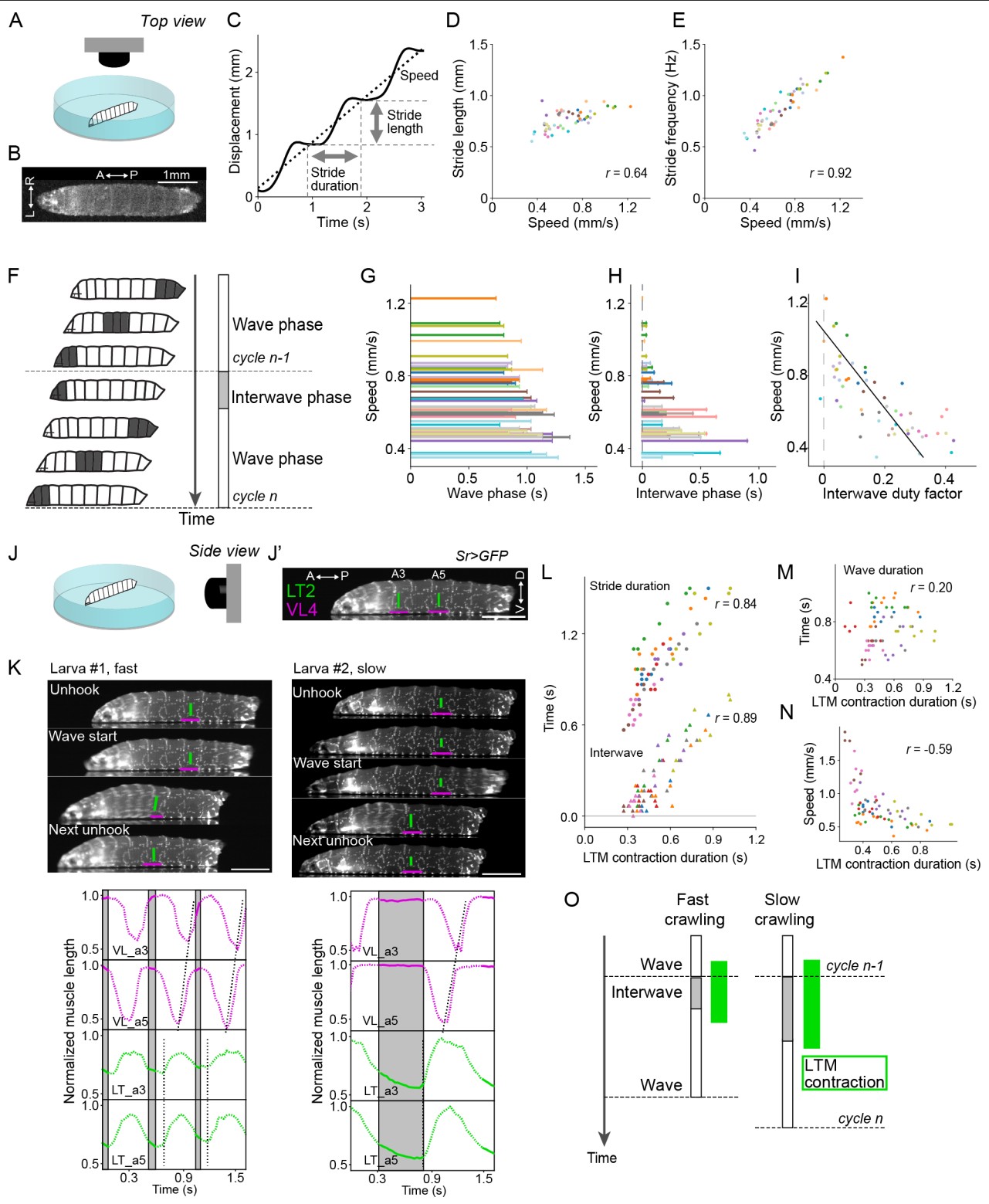

**Figure 1.** Crawling speed depends on the duration of interwave phase, during which the lateral transverse (LT) muscles are contracted. (**A–I**) Recording of locomotion parameters from top view (n = 268 strides, 54 episodes, 18 larvae). (**A**) Schematic drawing of the crawling assay from top view. (**B**) An example frame of top-view recording. (**C**) Measurement of the stride duration, stride length, and speed. (**D**) Relationship between speed and mean stride length. Each dot shows a single episode. *r* represents Pearson correlation coefficient. (**E**) Relationship between speed and mean stride frequency. Each dot shows a single episode. (**F**) Schematic representation of the two phases of a locomotor cycle. (**G**) Relationship between the duration of wave

*Figure 1 continued on next page*

*Figure 1 continued*

phase and speed. (**H**) Relationship between the duration of interwave phase and speed. (**I**) Interwave duty factor, the proportion of interwave phase in the stride duration, decreases with speeds. Linear regression coefficient is –0.47. (**J–O**) Recording of locomotion parameters and muscular kinematics from side view (n = 9 larvae, 18 episodes, 67 strides). Scale bars: 1 mm. (**J**) Schematic drawing of the crawling assay from side view. (**J'**) An example frame of side-view recording. LT2: lateral transverse muscle 2; VL4: ventral longitudinal muscle 4. (**K**) Representative tracking of the muscle movement during forward crawling. Top-left panel shows the muscle movement with a fast speed. Top-right panel shows the muscle movement with a slow speed. Bottom panels demonstrate the dynamics of muscle lengths in the data shown in the top panels. (**L**) Relationship between the contraction duration of LT2 muscle and two temporal parameters (circles indicate stride duration, and triangles indicate interwave duration) (Pearson correlation coefficients; stride duration: 0.84 and interwave duration: 0.89). (**M**) Relationship between the contraction duration of LT2 muscle and wave duration (Pearson correlation coefficient: 0.20). (**N**) Relationship between the mean duration of LT2 muscle contraction and speed. (**O**) Schematic of the relationship between LT muscle (LTM) contraction and crawling speed. The duration of the two phases and the contraction of LT muscles are correlated with crawling speed. In (**D, E, G, H, I, L, M, N**), different colors indicate different animals.

The online version of this article includes the following video and figure supplement(s) for figure 1:

**Figure supplement 1.** Kinematics of larval crawling and muscle length.

**Figure 1—video 1.** Side-view imaging video of a larva showing fast crawling.

https://elifesciences.org/articles/83328/figures#fig1video1

**Figure 1—video 2.** Side-view imaging video of a larva showing slow crawling.

https://elifesciences.org/articles/83328/figures#fig1video2

The phase-specific contraction of transverse muscles led us to investigate their role during the interwave phase. Accordingly, we analyzed the contraction duration of transverse muscles and analyzed its relationship with the duration of the two locomotor phases. While the duration of the wave phase was not strongly correlated, the duration of the interwave phase had a high correlation with the contraction duration of transverse muscles (wave phase $r$ = 0.20, interwave phase $r$ = 0.89, p<0.0001; the Pearson correlation coefficient, *Figure 1L and M*). Stride duration, which is the sum of the wave duration and the interwave duration, also had a strong correlation with the contraction duration of transverse muscles ($r$ = 0.84, *Figure 1L*). This result suggests that the contraction duration of transverse muscles could be related to crawling speed. We therefore plotted the duration of transverse muscle contractions against crawling speed (*Figure 1N*) and found that they were indeed correlated ($r$ = –0.59). Next, we analyzed the relationship between the amplitude of transverse muscle contraction and the crawling kinematics. As is the case with the contraction duration, the contraction amplitude is also correlated with the interphase duration and stride duration but not the wave duration (*Figure 1—figure supplement 1H and I*). On the other hand, the contraction amplitude is only weakly correlated with speed (*Figure 1—figure supplement 1J*). These results show that the duration and amplitude of the synchronous contractions of transverse muscles are related to the duration of interwave phase (*Figure 1O*). Importantly, the duration of the synchronous contraction is correlated with crawling speed (*Figure 1O*).

## Identification of GABAergic interneurons A31c showing segmentally synchronized activity

In a screen for neurons that are activated during the interwave phase, we identified cell-type A31c (*Figure 2—figure supplement 1*). By reviewing the existing genetic driver expression patterns (*Li et al., 2014*), we identified several genetic drivers targeting the A31c neuron, including a split GAL4 driver (*A31c-a8-sp*) specifically targeting the A31c neuron in segment A8, a split GAL4 driver (*A31c-sp*) targeting A31c neurons in neuromeres A2-A8, and a LexA driver (*A31c-LexA*) that targets A31c neurons in neuromeres A2-A8 with variable expression patterns. We first used these lines to investigate the morphology and neurotransmitter identity of A31c neurons. The neurites project dorsally by approximately one neuromere, mostly to the anterior (*Figure 2A*). The synaptic input sites are located along the dorsolateral (DL) tract (*Landgraf et al., 2003*), while the synaptic output sites are mainly positioned dorsally near the midline (*Figure 2A and B*). Using immunohistochemistry, we found that A31c neurons are GABAergic (*Figure 2—figure supplement 1A*).

We then used a dual-color imaging system to monitor the activity of A31c neurons using *A31c-sp>UAS-CD4::GCAMP6f* and the pan-neuronal activity using *nSyb-LexA>LexAop-RGECO1* in the isolated CNS (*Figure 2C–E* and *Figure 2—figure supplement 1B and C*; see 'Materials and methods' for details). The pan-neuronal activity patterns were used as an indicator of the fictive behaviors

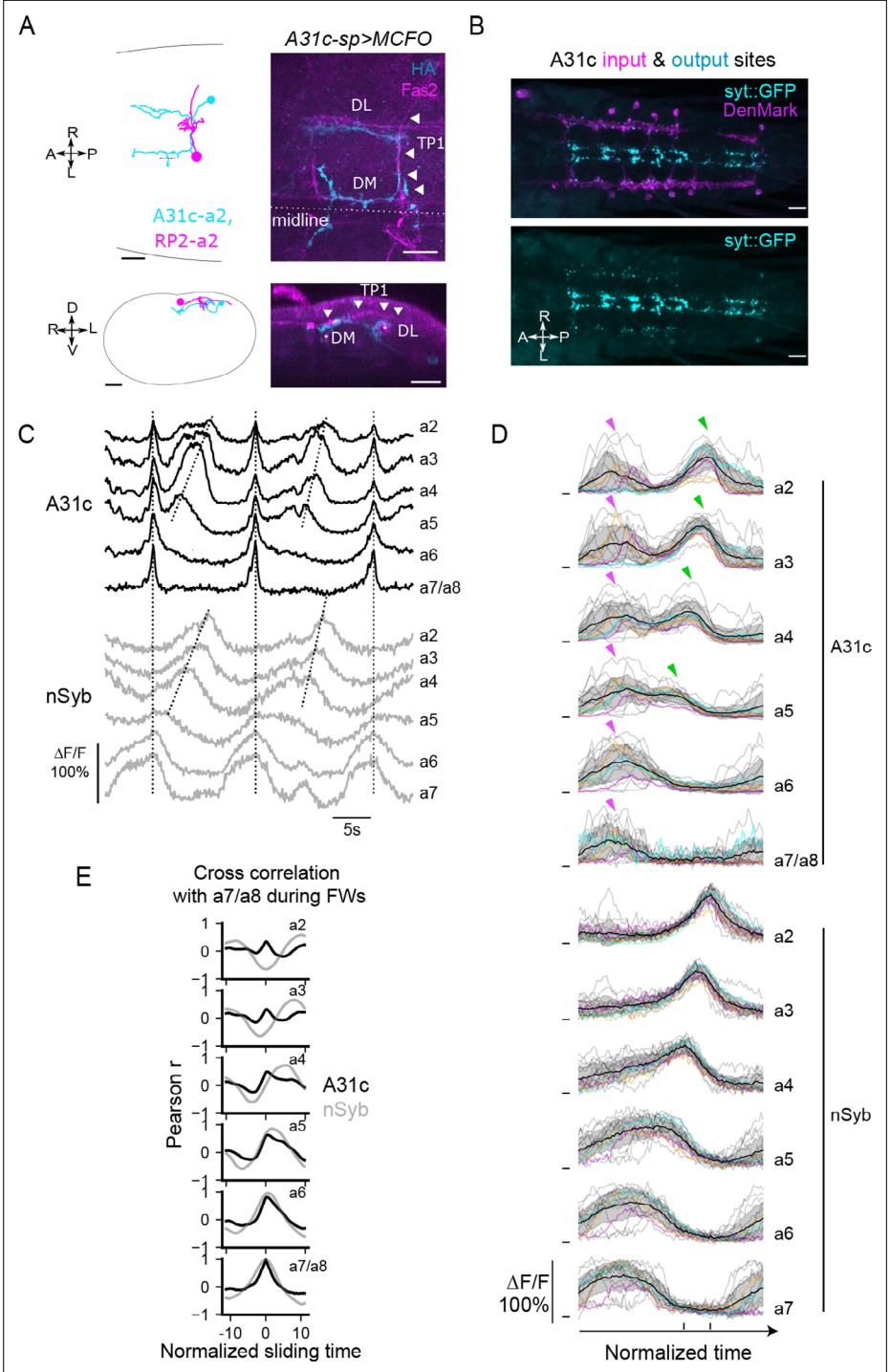

**Figure 2.** A31c neurons show synchronous activity preceding the forward wave. (**A**) A31c single-neuron morphology shown by EM reconstruction and genetically mosaic analysis. TP1: transverse projection 1; DM: dorsomedial fascicle; DL: dorsolateral fascicle (*Landgraf et al., 2003*). Arrowheads indicate the tract of TP1. Scale bars: 20 µm. (**B**) Pre- and postsynapse markers label the input and output sites of A31c neurons (*A31c-sp>UAS-syt::GFP, UAS-DenMark*). Scale bars: 20 µm. (**C–E**) Recording of calcium activity of A31c neurons (*A31c-sp>UAS-CD4::GCaMP6f*) and group activity of nSyb neurons (*nSyb-LexA>LexAop-RGECO1*) which reports the pan-neuronal activity in fictive locomotion (n = 5 larvae, 15 traces). (**C**) Example recordings of A31c neurons and nSyb neurons in fictive forward locomotion. The vertical dashed lines indicate synchronous activity. The inclined dashed lines indicate neural activity that corresponds to fictive forward crawling. (**D**) Group data of calcium imaging of A31c neurons and nSyb neurons. Each trace is aligned by activity peak of nSyb_a4 and nSyb_a2 and normalized to 0–1

*Figure 2 continued on next page*

*Figure 2 continued*

by the activity maximum and minimum of the whole recording. Magenta arrows indicate the co-activation of A31c neurons. Green arrows indicate the wave-like activity of A31c neurons. Black lines represent the average calcium activity. Shading represents the standard error. Colored lines represent the three example traces. Gray lines represent all other traces. Ticks along the horizontal axis indicate the activity peaks of nSyb_a4 and nSyb_a2. Ticks along the vertical axis indicate the 0. (**E**) Cross-correlation of neuronal activity between the neuron in each segment (from A2 to A7/A8) and the one in A7/A8 (black: A31c neurons; gray: nSyb neurons). See 'Materials and methods' for details.

The online version of this article includes the following figure supplement(s) for figure 2:

**Figure supplement 1.** The neurotransmitter identity and calcium activity of A31c neurons.

produced (*Lemon et al., 2015*), including stereotyped fictive forward waves (FW). At the initiation of forward locomotion, all abdominal A31c neurons show burst-like coactivation preceding the FW (*Figure 2C and D*). As observed in muscle contraction patterns described in the previous section (*Figure 1—figure supplement 1G'*), panneuronal signals in the posterior-most segments (A6 and A7) show preparatory synchronous activity (*Figure 2C*), which causes the overlap of A31c activity with the posterior segmental panneuronal activity. However, the synchronous activity of A31c neurons primarily occurred between FW propagations. During the FW that follows, A31c neurons in anterior segments A2-A5 are re-activated in a wave-like sequence, which could indicate their potential involvement in peristaltic waves (*Figure 2C and D*). The intersegmental lags of pan-neuronal activity between neighboring segments show non-zero values, which reflects the propagation of neuronal activity along the body axis (*Figure 2E*). On the other hand, the intersegmental lag of A31c activity during the interwave phase is near zero, consistent with their synchronized activity (*Figure 2E*). To sum, these results show that A31c neurons exhibit synchronous multi-segmental activity during the interwave phase.

## A31c neurons receive synaptic inputs from descending neurons and provide synaptic output to local and ascending neurons

To understand the details of the connectivity of the circuit A31c is part of, we reconstructed the connectivity of A31c using EM connectomics (see 'Materials and methods' for details). We identified A31c neurons in neuromeres A2-A8 in the database of the larval CNS, reconstructed all pre- and postsynaptic partners, and analyzed their connectivity (*Figure 3A–C* and *Figure 3—figure supplement 1A and B*). We analyzed the connectivity of A31c neurons in anterior segments A2-A3 and posterior segments A7-A8 separately (*Figure 3B–D*). We found that the synaptic inputs to A31c neurons are similar in the anterior and posterior segments, with several descending cell types innervating A31c across segments (*Figure 3D*). The same subesophageal descending neuron cell type (here labeled 'S10') provides a significant plurality of the synaptic input. Among postsynaptic targets, we found that just one cell type is consistent between segments A2-A3 and A7-A8: A26f neurons, which are among their top three postsynaptic partners (*Figure 3D*). A26f neurons also receive synaptic inputs from the ascending cell type A19f, one of the top postsynaptic partners of A31c neurons (*Jonaitis, 2020*). Interestingly, it has previously been reported that A26f strongly innervates the transverse motor neurons (*Zarin et al., 2019*; *Zwart et al., 2016*).

We next used *trans*-Tango, a genetic tool for tracing postsynaptic partners (*Talay et al., 2017*), to confirm the identity of the postsynaptic neurons of A31c-a8. We repeatedly identified Tango expression in an A26f-like cell type in segment A7, in addition to other neurons, some of which we could identify (four samples showing A26f-a7 neurons; *Figure 3—figure supplement 1C and D*). These results collectively show that A26f neurons are postsynaptic to A31c neurons.

## A26f GABAergic premotor neurons show segmentally synchronized activity preceding the fictive forward wave

Since A26f neurons are postsynaptic to A31c (*Figure 3D* and *Figure 3—figure supplement 1C and D*) and strongly innervate LT motor neurons (*Figure 4A* and *Figure 4—figure supplement 1A*; *Zarin et al., 2019*; *Zwart et al., 2016*), we next focused on A26f neurons to understand the neural mechanism underlying the generation of the interwave phase. We used a split GAL4 driver ('*A26f-sp*'), which labels A26f neurons in neuromeres A3-A5 (*Figure 4B*), to investigate their morphology and

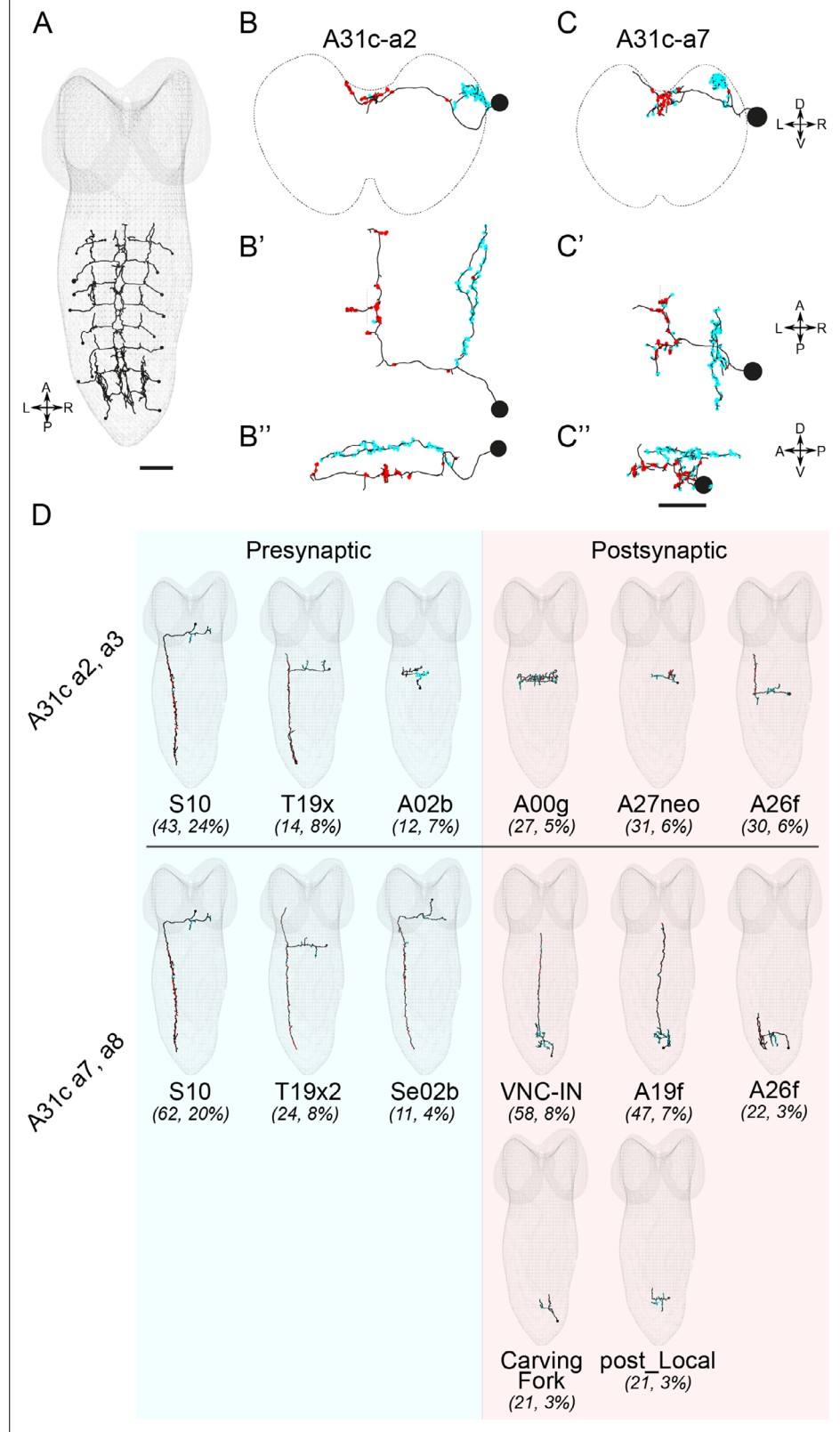

**Figure 3.** EM reconstruction reveals the connectivity of A31c neurons. (**A**) Identification of all A31c neurons in the EM database. Scale bar: 20 μm. (**B–B'' and C–C''**) Morphology of an anterior (**B–B''**) and a posterior (**C–C''**) A31c neuron. (**B, C**) Frontal view, (**B', C'**) dorsal view, and (**B'', C''**) lateral view. Red circles indicate the output sites. Cyan

*Figure 3 continued on next page*

*Figure 3 continued*

circles indicate the input sites. Scale bar: 10 μm. (**D**) Top pre- and postsynaptic partners of anterior and posterior A31c neurons.

The online version of this article includes the following figure supplement(s) for figure 3:

**Figure supplement 1.** Connectivity of A31c neurons revealed by EM reconstruction and trans-synaptic labeling.

neurotransmitter identity. A26f neurons have synaptic input sites dorsally near the midline and synaptic output sites near the DL tract (*Figure 4B*). Remarkably, A26f neurons project their axon along the DL tract for multiple segments. As a representative example, the axon of A26f in neuromere A5 extends four neuromeres from A6 neuromere to A3 neuromere (*Figure 4B*). We found that A26f neurons are GABAergic (*Figure 4—figure supplement 1B*). It has previously been reported that A26f neurons are corazoninergic (*Zarin et al., 2019*). However, a comparison of the morphology of A26f neurons with confirmed corazoninergic neurons (*Santos et al., 2007*) and the absence of peptidergic dense core vesicles in A26f neurons in the EM connectomics dataset suggest that A26f neurons are not corazoninergic. A26f neurons form inhibitory synapses onto MNs innervating LT muscles in multiple neuromeres, which suggests the potential of A26f neurons to control the activity of LT muscles broadly in multiple segments.

Next, we related the activity patterns of A26f neurons to fictive crawling by performing dual-color imaging experiments of *A26f-sp>UAS-CD4::GCaMP6f* and *nSyb-LexA>LexAop-RGECO1* in the isolated CNS (*Figure 4C* and *Figure 4—figure supplement 1C and D*; see 'Materials and methods' for further details). This line restricts expression to A26f in segments A3-A5. Unlike most neurons, which show fictive wave-like activity (*Lemon et al., 2015*), the A26f neurons we could image with this line only showed synchronized activity, which mostly occurred before the fictive peristaltic wave progressed anteriorly from segment A4 (*Figure 4C and D*). Instead, A26f activity overlaps considerably with the synchronous increases in activity seen in the panneuronal line in segments A7-A5 that characterize the interwave phase (*Figure 4C*; see 'Discussion'). Consistent with their synchronized activity, the activity of A26f segmental homologs is highly correlated, unlike pan-neuronal activity (*Figure 4E*). A26f neurons can exhibit one or several peaks at the initiation phase of the FW (*Figure 4C and D*). To sum, A26f neurons have four important characteristics: (1) they form inhibitory synapses with motor neurons in multiple segments targeting transverse muscles; (2) A26f neurons in the abdominal segments are activated simultaneously; (3) A26f neurons are activated between wave phases; and (4) they are postsynaptic to A31c neurons.

As both A26f and A31c neurons show robust synchronous activity at the initiation of FW, we then monitored the activity of the two neurons simultaneously by using *A31c-LexA>LexAop-jRGECO1b, A26f-sp>UAS-CD4::GCaMP6f*. We found that the synchronous peak of A26f neurons is 'bookended' by the peaks in A31c activity of neighboring segments (*Figure 4F*). This alternating activity pattern between A31c and A26f is consistent with the inhibitory nature of the A31c-A26f synapses and suggests these cell types might be involved in determining the duration of the interwave phase.

## Activation of A26f neurons reduces the amplitude of LT muscle contractions during forward crawling

To assess whether A26f neurons can inhibit the activity of LT muscles, we analyzed muscle responses to the optogenetic activation of A26f neurons during forward peristaltic waves. We combined the optogenetic activator *UAS-CsChrimson* targeted by *A26f-sp* to activate A26f neurons and the muscle genetic marker *mhc-GFP* expressing GFP to visualize the body wall muscles (*A26f-sp>UAS-CsChrimson, mhc-GFP*). We used *A26f-sp* negative animals as a control (*UAS-CsChrimson, mhc-GFP*). Because of the spectral overlap between the light to activate CsChrimson and that to excite GFP, we used a confocal microscopy system that separates the light for optogenetics and imaging into two sections of the objective back aperture, respectively, in combination with a new preparation we call 'sideways preparation' (*Figure 5A and B*; see 'Materials and methods' for detail).

We tracked the length of muscle LT2 and longitudinal muscle VL2 in segment A5 upon optogenetic stimulation. Activation of the A26f neurons reduced the contraction amplitude of the LT2 muscle, while the contraction of the VL2 muscle was almost unchanged (*Figure 5C and D* and *Figure 5—figure*

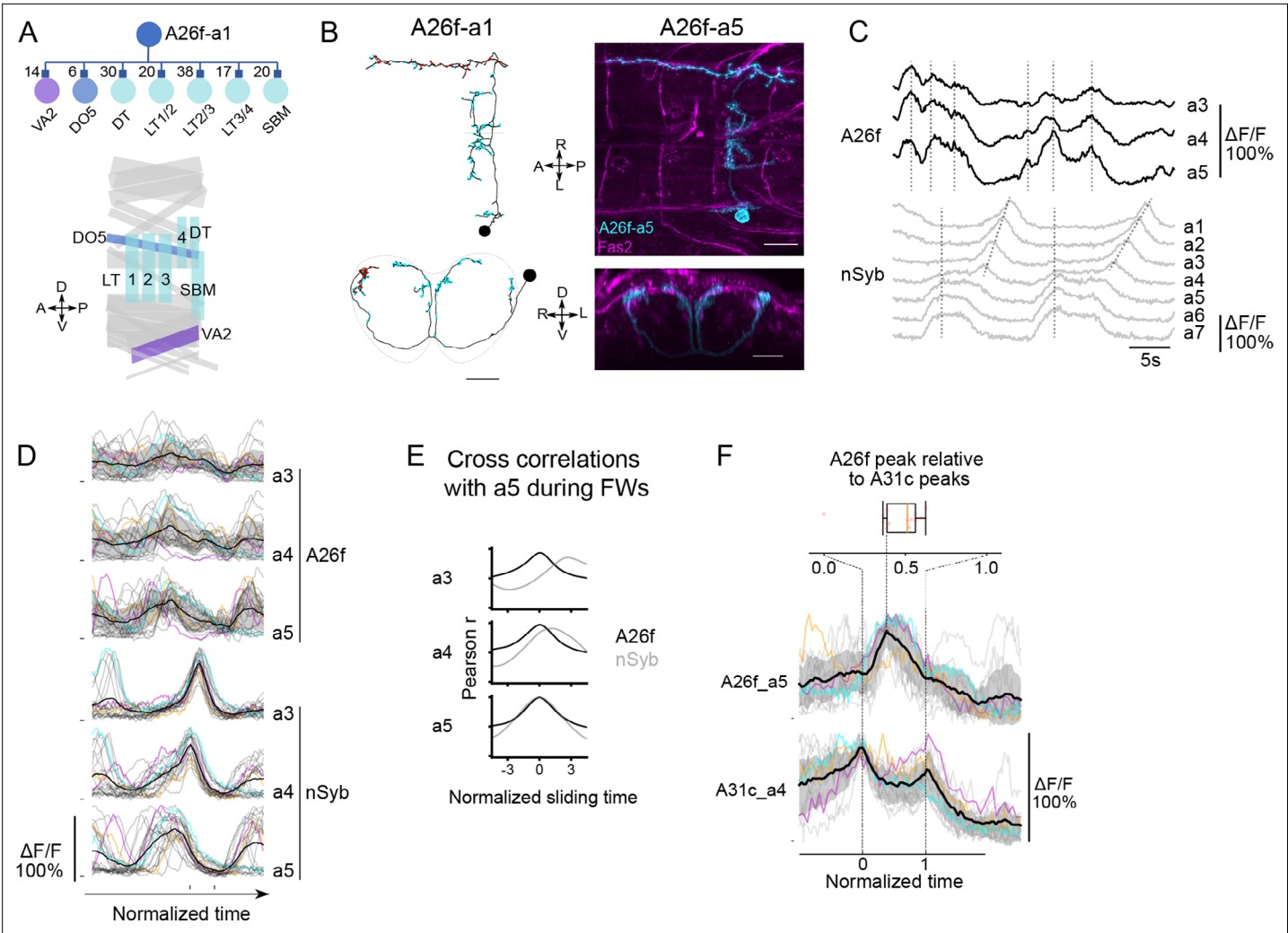

**Figure 4.** A26f neurons inhibit motor neurons and exhibit synchronous activity at the initiation of forward locomotion. (**A**) A26f neurons innervate motor neurons. Top: postsynaptic neurons of A26f neurons revealed by connectomics analysis in A1 neuromere (*Zarin et al., 2019*). Bottom: layout of body wall muscles in a hemi-segment. Purple, blue, and sky-blue muscles are innervated by motor neurons in the same color in the top panel. (**B**) Morphology of A26f neurons shown by the EM reconstruction and confocal images. Scale bars: 10 μm. (**C–E**) Recording of calcium activity of A26f neurons (*A26f-sp>UAS-CD4::GCaMP6f*) and group activity of nSyb neurons (*nSyb-LexA>LexAop-RGECO1*) (n = 6 larvae, 18 traces). (**C**) Example recordings of A26f neurons and nSyb neurons in fictive forward locomotion. The vertical dashed lines in the A26f plot indicate synchronous activity. The vertical dashed lines in the nSyb traces indicate synchronous activity at the posterior-most segments during the interwave phase. The inclined dashed lines in the nSyb traces indicate neural activity that corresponds to fictive forward crawling. (**D**) Group data of calcium imaging of A26f neurons and nSyb neurons. Each trace is aligned to the activity peak of nSyb_a4 and nSyb_a2 and normalized to 0–1 by the activity maximum and minimum of the whole recording. Black lines represent the average calcium activity. Shading represents the standard error. Colored lines represent the three example traces. Gray lines represent all other traces. Ticks along the horizontal axis indicate the activity peaks of nSyb_a4 and nSyb_a2. Ticks along the vertical axis indicate the 0. (**E**) Cross-correlation of neuronal activity between the neuron in each segment (A3–A5) and the one in A5 (black: A26f neurons, gray: nSyb neurons). See 'Materials and methods' for details. (**F**) Simultaneous calcium imaging of A31c and A26f neurons. Bottom: recording of calcium activity of A26f-a5 neuron (*A26f-sp>UAS-CD4::GCaMP6f*) and A31c-a4 neuron (*A31c-LexA>LexAop-jRCaMP1b*) (n = 6 larvae, 13 traces). Colored lines indicate the example traces. Black lines indicate the average calcium activity. Gray lines indicate all other traces. The left vertical dashed line indicates the first synchronous activity of A31c (normalized time = 0), and the right vertical dashed line indicates the second synchronous activity of A31c (normalized time = 1). Top: peak time of A26f signals relative to the first peak time of A31c signals.

The online version of this article includes the following figure supplement(s) for figure 4:

**Figure supplement 1.** LT motor neuron connectivity and the neurotransmitter identity and calcium activity of A26f neurons.

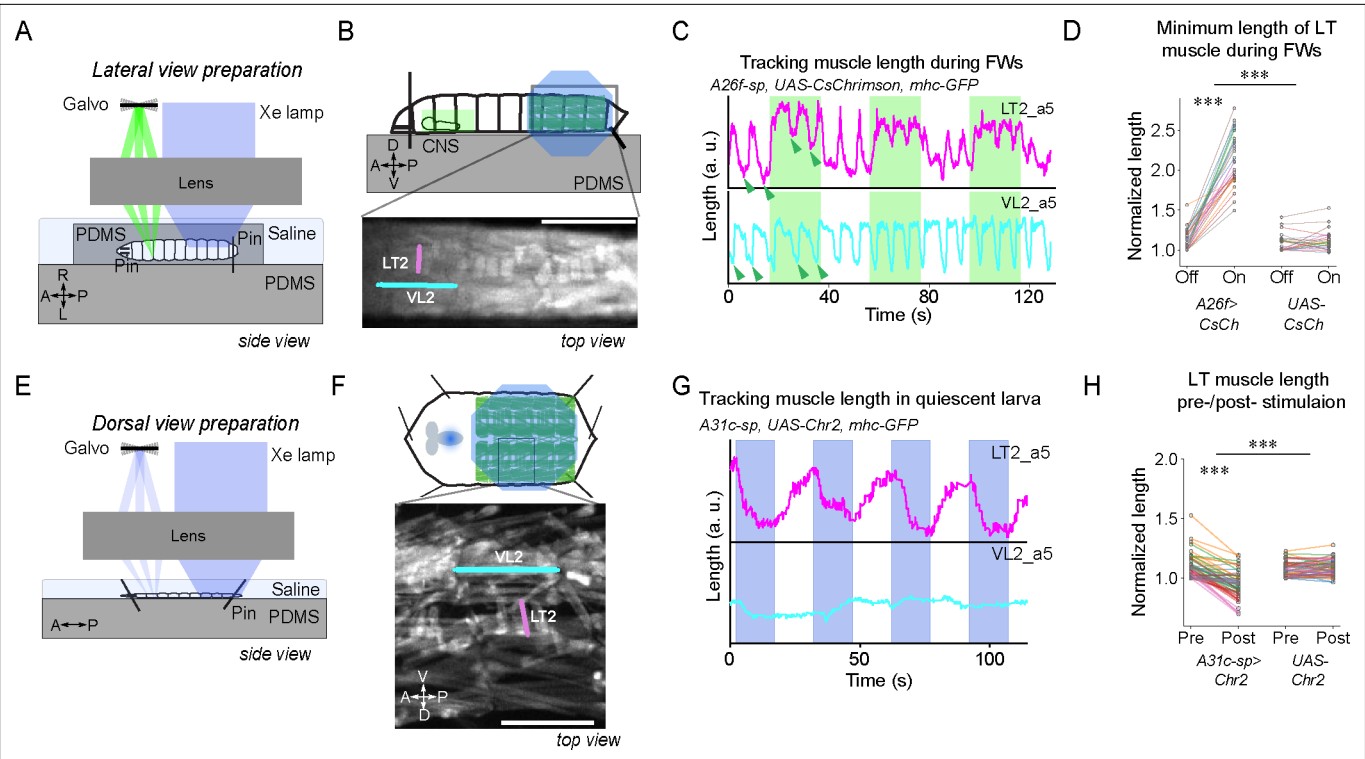

**Figure 5.** Optogenetic activation of A31c or A26f neurons affects the contraction of lateral transverse (LT) muscles. (**A–D**) Optogenetic activation of A26f neurons reduces the contraction amplitude of the LT2 muscle during forward crawling in the sideways preparation. (**E–H**) Optogenetic activation of A31c neurons causes contraction of the LT muscle in the fillet preparation. (**A, E**) Experimental setups. See 'Materials and methods' for details. (**B, F**) Schematics of the imaging setup (top) and sample fluorescence images (bottom). Scale bars: 0.5 mm. (**C, G**) Traces of the length of the transverse muscle LT2 and the longitudinal muscle VL2 in the optogenetic experiments. Shaded regions show the timing when the light stimulus is applied. Arrowheads indicate where the measurement was made in *Figure 5D*. (**D**) The minimum length of the LT2 muscle was increased by the activation of A26f neurons (A26f group: n = 8 larvae, 28 trials; control group: n = 8 larvae, 28 trials). Muscle lengths are normalized to the minimum length during the light-off period. Different colors indicate different animals. The hierarchical bootstrap test is used (see 'Materials and methods' for details). (**H**) The length of the LT2 muscle in quiescent larvae was decreased by the activation of A31c neurons (A26f group: n = 8 larvae, 62 trials; control: n = 8 larvae, 56 trials). Muscle lengths are normalized to the minimum length during the light-off period. Different colors indicate different animals. The hierarchical bootstrap test is used (see 'Materials and methods' for details).

The online version of this article includes the following figure supplement(s) for figure 5:

**Figure supplement 1.** The length of longitudinal VL muscles upon optogenetic activation of A26f and A31c neurons.

---

*supplement 1A*). These results confirm that activation of A26f neurons is sufficient for the inhibition of the contraction of LT muscles.

## Activation of A31c neurons induces the contraction of LT muscles

As A31c could inhibit A26f based on its connectivity (*Figure 3D* and *Figure 2—figure supplement 1A*) and the activity analyses described above (*Figure 4F*), we tested whether activation of A31c neurons can enhance contractions of LT muscles. We therefore activated A31c neurons and analyzed the change in LT2 muscle length using animals carrying *A31c-sp>UAS-Chr2.T159C, mhc-GFP* transgenes. We restricted the stimulation laser to the abdominal neuromeres in a semi-intact preparation ('fillet preparation,' *Lemon et al., 2015*) to avoid activating SEG or brain neurons (*Figure 5E and F*). The stimulation caused the contraction of LT muscles in all visualized abdominal neuromeres (A3-A8; *Figure 5G*), causing a reduction in the minimum length of the LT2 muscle (*Figure 5H*). These results suggest that activation of A31c neurons is sufficient to activate the LT muscles. As no apparent contraction of other muscles was observed (*Figure 5—figure supplement 1B*), we assume that the A31c neurons mainly regulate the activity of the LT muscles.

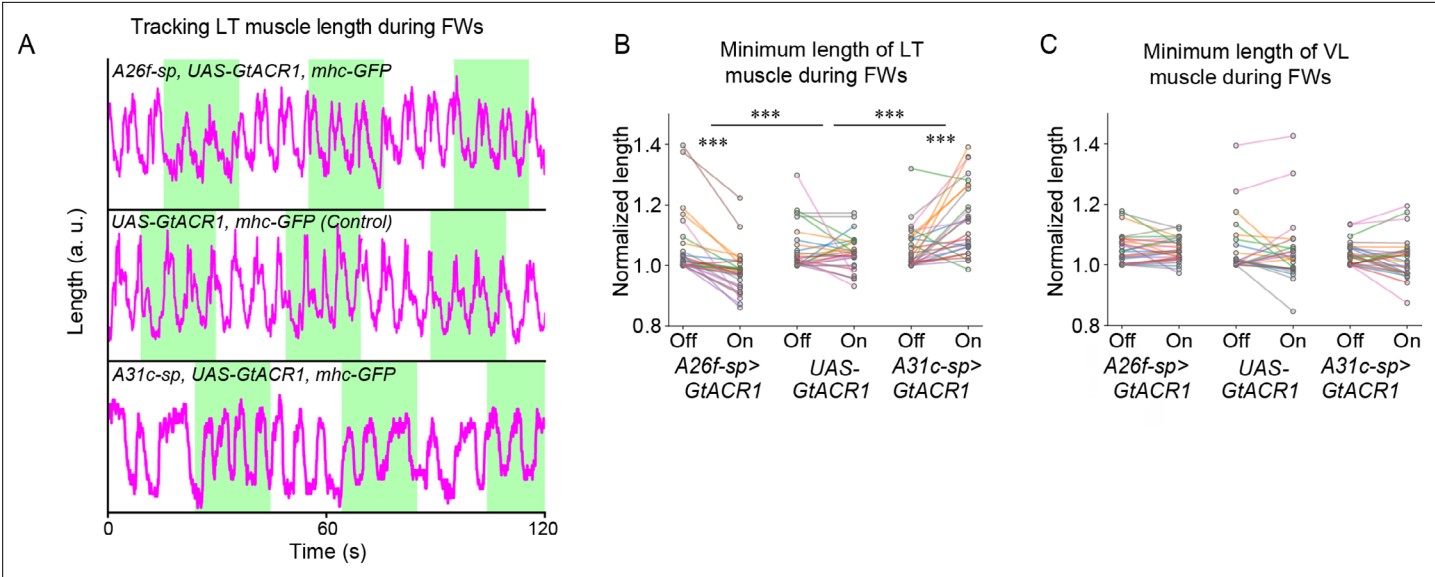

**Figure 6.** Optogenetic inhibition of A31c or A26f neurons affects the contraction of lateral transverse (LT) muscles. (**A**) Traces of the length of transverse muscles (LT2) in the sideways preparation with optogenetic stimulation. (**B, C**) Quantification of the minimum length of LT muscles and VL muscles by the optogenetic inhibition of A26f or A31c neurons (A26f group: n = 8 larvae, 28 trials; control group: n = 8 larvae, 26 trials; A31c group: n = 8 larvae, 27 trials). Muscle lengths are normalized to the minimum length during the light-off period. Different colors indicate different animals. The hierarchical bootstrap test is used (see 'Materials and methods' for details). (**B**) The minimum length of LT muscles was affected by the optogenetic inhibition of A26f or A31c neurons. (**C**) VL muscles was not affected by the optogenetic inhibition of A26f or A31c neurons.

## Silencing A31c or A26f neurons influences the amplitude of LT muscle contractions during forward crawling

Next, we examined whether the interneurons of interest were required for the observed contraction of the LT muscles. To test this, we used optogenetic silencing combined with muscular imaging in the sideways preparation (*Figure 5A*). We first tested whether A26f neurons are required for the contractions of transverse muscles by using animals carrying *A26f-sp>UAS-GtACR1, mhc-GFP* for optogenetic silencing and muscular visualization. We found that the minimum length of muscle LT2 decreased after optogenetic silencing of A26f, suggesting increased levels of contraction (*Figure 6A and B*). We next assessed the requirement of A31c neurons by using *A31c-sp>UAS-GtACR1, mhc-GFP*. We found that after optogenetic silencing the minimum length of the LT2 muscle in segment A5 was increased during forward cycles (*Figure 6A and B*). In contrast, the minimum length of muscle VL2 was not affected by the inhibition of A26f or A31c (*Figure 6C*). These results reveal that the activity of A26f and A31c neurons is necessary for the appropriate contractions of LT muscles observed during locomotor cycles.

## A26f neurons modulate interwave duration

Our previous results suggest that the activation of A26f neurons reduces the contraction of the LT muscles, thereby potentially reducing the duration of the interwave phase. To test this hypothesis, we activated A26f neurons and analyzed the kinematics of crawling in animals of the genotype *A26f-sp>CsChrimson* on low-concentration agarose plates (0.7%; *Figure 7A and B*). We used animals that lacked the *A26f.DBD* transgene as a control (*A26f.AD>CsChrimson*).

During optogenetic activation of A26f neurons, larvae exhibited faster crawling (*Figure 7B*). By analyzing the kinematics, we confirmed that the interwave phase and the total stride duration were both significantly decreased during the optogenetic activation (*Figure 7C and C'*). On the other hand, the wave duration was slightly increased (*Figure 7—figure supplement 1A*). Consistent with these results, the speed of crawling was significantly increased (*Figure 7C''*). These results show that the activation of A26f neurons leads to an increase in stride frequency and speed.

Next, we asked whether A26f neurons are required to regulate the interwave phase and thereby the speed of freely crawling animals. To this end, we optogenetically inhibited A26f neurons in animals

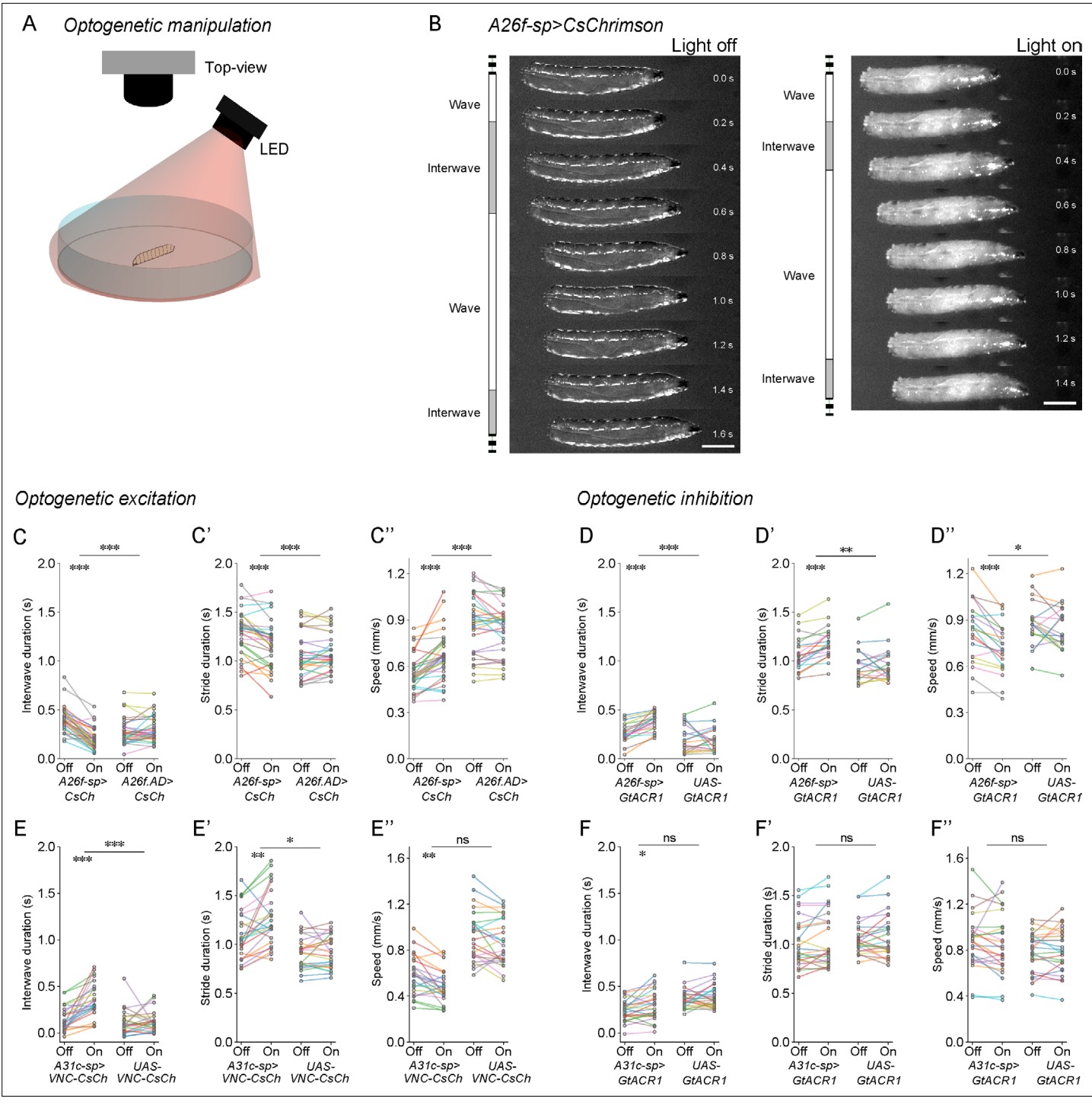

**Figure 7.** Optogenetic manipulation of A26f neurons affects the interwave duration. (**A**) Experimental setup for optogenetics in free-crawling larvae. (**B**) Example frames show that the interwave phase is reduced during the activation of A26f neurons. Scale bars: 1 mm. (**C–C''**) Crawling dynamics changed during the activation of A26f neurons (A26f group: n = 10 larvae, 30 trials; control group: n = 10 larvae, 30 trials). (**D–D''**) Crawling dynamics changed during the inhibition of A26f neurons (A26f group: n = 10 larvae, 20 trials; control group: n = 10 larvae, 20 trials). (**E–E''**) Crawling dynamics changed during the activation of A31c neurons (A31c group: n = 10 larvae, 25 trials; control group: n = 10 larvae, 25 trials). (**F–F''**) Crawling dynamics changed during the inhibition of A31c neurons (A31c group: n = 10 larvae, 25 trials; control group: n = 10 larvae, 25 trials). In (**C–F''**), different colors indicate different animals and the hierarchical bootstrap test is used (see 'Materials and methods' for details).

The online version of this article includes the following figure supplement(s) for figure 7:

**Figure supplement 1.** Analysis of crawling kinematics of larvae with optogenetic perturbation.

**Figure supplement 2.** Confocal images showing the expression patterns of split GAL4 drivers in the central nervous system (CNS).

carrying *A26f-sp>GtACR1* and analyzed their crawling kinematics (*Figure 7D–D"* and *Figure 7—figure supplement 1C and D*). We found that inhibiting the A26f neurons increased the interwave duration and stride duration but had no significant effect on the wave duration (*Figure 7D and D'* and *Figure 7—figure supplement 1C*), resulting in decreased speeds (*Figure 7D"*). Combined with our previous analyses, these results indicate that the A26f neurons are involved in regulating the speed of locomotion by modulating the contraction of LT muscles.

## Activation of A31c neurons caused the increases in interwave duration and stride duration

Next, we tested the effect of manipulating A31c on crawling (*Figure 7E and F* and *Figure 7—figure supplement 1E–H*). To activate A31c neurons, we used a genetic system *UAS-VNC-CsChrimson* that confines the expression of CsChrimson to the VNC neurons targeted by the *A31c-sp* transgene, resulting in expression in neuromeres A2-A8. We used animals carrying *UAS-VNC-CsChrimson* only as a control. During the activation of A31c neurons, the interwave duration and the stride duration were significantly increased, while no significant difference was found in the wave duration (*Figure 7E and E'* and *Figure 7—figure supplement 1E*). These effects are consistent with those observed during the inhibition of A26f neurons (*Figure 7D and D'* and *Figure 7—figure supplement 1C*). On the other hand, the inhibition of A31c neurons, which we achieved by expressing *GtACR1* in A31c neurons using the *A31c-sp* driver line, did not induce defects in crawling kinematics (*Figure 7F–F"* and *Figure 7—figure supplement 1G and H*). This implies that additional cell types to A31c could set the level and timing of activity in A26f and therefore the speed of locomotion. Moreover, *A31c-sp>GtACR1* is expressed in cells in the brain and subesophageal ganglion. However, the majority of *A31c-sp+* cells in the brain are positive to Deadpan (dpn) (*Figure 7—figure supplement 2D*), a marker for neuroblast (*Doe, 2017*), and the terminals of *A31c-sp+* cells in the brain lack synapse-like morphology, suggesting that most *A31c-sp+* cells in the brain are not mature neurons. Our data therefore imply that A31c neurons could contribute to the regulation of interwave phase duration, possibly through

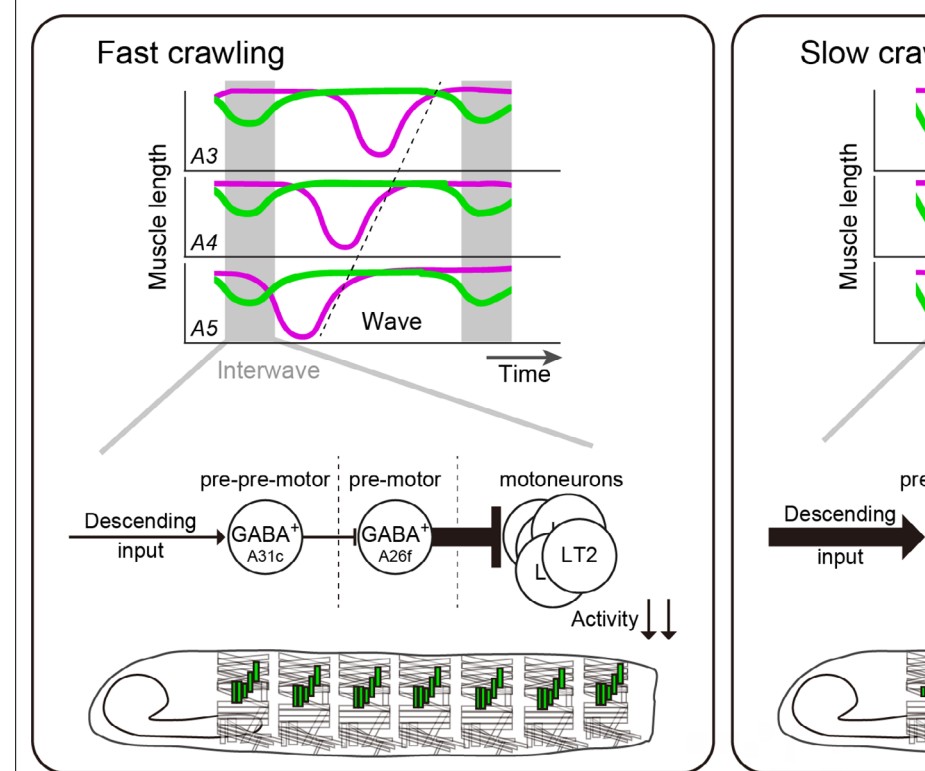
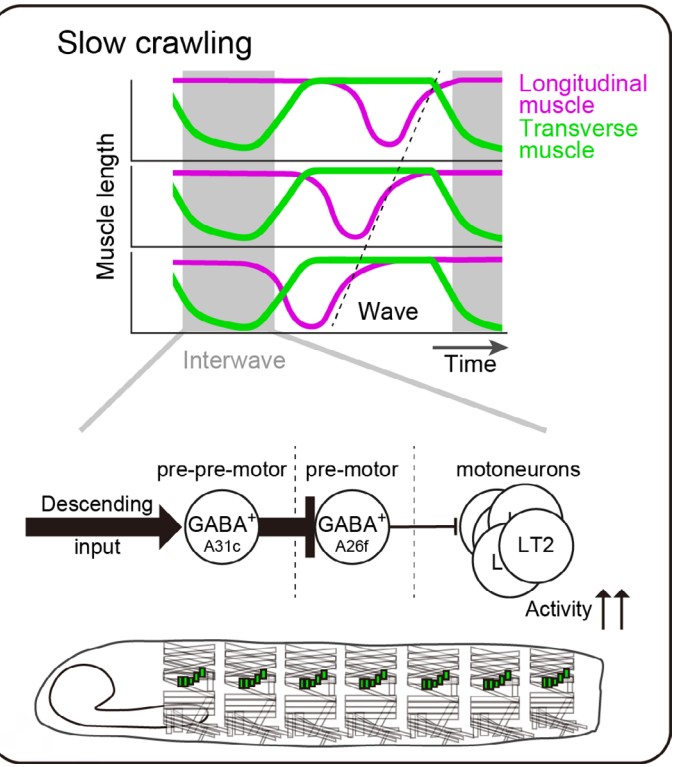

**Figure 8.** Schematics of larval speed control by A26f and A31c. Contraction of transverse muscles is suppressed by A26f neurons to make the interwave phase short and crawl fast (left panel). Multi-segmental synchronous activity of A31c and A26f neurons elongates the interwave phase duration to decrease crawling speed (right panel).

A26f neurons, by the multi-segmental synchronous excitation of transverse muscles within the interwave phase.

## Discussion

In summary, we aimed to understand the neural mechanisms that underlie the selective modulation of one phase of the locomotor cycle during speed control. We used the *Drosophila* larva as a model and found that this animal uses a strategy to primarily vary the phase between consecutive peristaltic waves for speed regulation. To implement this strategy, the larva modulates the amplitude and duration of the contraction of the LT muscles that are perpendicular to the crawling direction, which contract synchronously along the anterior-posterior axis before the onset of the peristaltic wave. The GABAergic interneurons A26f and A31c, upstream of the MNs innervating the LT muscles, showed segmentally synchronized activity preceding the FW. Connectivity analysis further revealed that A31c neurons receive shared descending input and synapse onto ascending neurons and local neurons, including A26f. These neurons are involved in modulating the activity of the LT muscles and thus the interwave duration and speed. Altogether, we established a neural basis for speed regulation by linking the speed-dependent modulation of contractions of muscles to the interneurons that control their activity (*Figure 8*).

### Mechanics and mechanisms of modulating the speed of locomotion

Locomotion speed is a function of stride frequency and stride length. We found that, similar to previous findings, the speed of *Drosophila* larval crawling is determined more so by stride frequency than stride length (*Frigon et al., 2014*; *Grillner et al., 1979*; *Jacobson and Hollyday, 1982*; *Nirody et al., 2021*). Furthermore, similar to limbed animals including mammals and other insects, the two constituent phases of a locomotor cycle vary differentially with speed, with the interwave phase varying more than the wave phase. This similarity between the dynamics of locomotion between limbed and axial locomotion could be indicative of the kinematic constraints of each type of movement. For limbed locomotion, the forces required to move the limb during the swing phase depend on the limb's mass. Large limbed animals, such as horses and humans (*Boije and Kullander, 2018*; *Minassian et al., 2017*), use momentum-based strategies, requiring only brief active contractions in swing muscles, whereas smaller animals such as stick insects and mice (*Bellardita and Kiehn, 2015*; *Bidaye et al., 2018*) require constant neural input onto the swing muscles. In both cases, the relative invariance of the duration of this phase suggests that the rotational inertia of swinging limbs may be an important limiting factor in limbed locomotion (*Kilbourne, 2013a*; *Kilbourne and Hoffman, 2013b*; *Kilbourne and Hoffman, 2015*; *Rocha-Barbosa et al., 2005*). The relative invariance of the peristaltic wave phase in the *Drosophila* larva suggests a similar constraint on the its motor system. Furthermore, the transverse muscles, and therefore its contractions during the interwave phase, may be an evolutionary adaptation, with an additional set of neurons having developed to control them. Consistent with this hypothesis, the last common ancestor to all bilaterians, the so-called 'Urbilaterian,' is thought to have only had circular and longitudinal muscles (*Cannon et al., 2016*), and transverse muscles have only been identified in the larvae of some other species of Diptera (e.g., *Tribolium castaneum*, *Schultheis et al., 2019*; *Galleria mellonella*, *Emery et al., 2019*). Furthermore, the transverse muscles are under independent neuromodulatory control (*Elliott et al., 2021*), and their motor neurons are innervated by a distinct set of interneurons (*Zwart et al., 2016*; *Kohsaka et al., 2019*).

In addition to the kinematic and evolutionary constraints, the differential modulation of the locomotor cycle may have several advantages. First, it could improve energy efficiency within a particular range of speeds: though the contraction and extension of LT muscles during the movement of the head must entail a metabolic cost, the energy cost of moving the center of mass (CoM) might be reduced during slower movements. The CoM is mainly moved in the pistoning phase during the head extension and the tail contraction (*Heckscher et al., 2012*), and the transverse muscle contractions, which we speculate are involved in driving head movements, might be a more efficient method of extending the anterior segments by regulating the hydrostatic skeleton (*Trimmer and Issberner, 2007*). This speculation is supported by our observation that in the posterior segments, longitudinal muscles contract during the interwave phase (as is also reflected in our activity imaging experiments in *Figures 2 and 4*, which show pronounced increases in activity in the posterior segments in between

fictive waves), which may act to further support the hydrostatic skeleton in driving head extension. Second, it could be a more efficient control strategy: independent control of the extension of the head and the peristaltic wave may reduce the complexity of motor control and increase the flexibility of the head and tail by allowing the two ends to be moved separately. This may be particularly important as the anterior-most segments are involved in other motor programs including feeding (*Melcher and Pankratz, 2005*), and the transverse muscles control self-righting behavior (*Picao-Osorio et al., 2015*).

## Neuronal control of speed modulation

A large body of work has identified the neural basis of the regulation of speed of locomotion in vertebrates, identifying associated circuits across the brain and spinal cord. Mouse, zebrafish, and *Xenopus* spinal cord preparations have been used to describe the selective recruitment of specific interneuron and motor neurons at different speeds (*Berg et al., 2018*; *Boije and Kullander, 2018*; *Gatto and Goulding, 2018*; *Grillner and El Manira, 2020*; *Grillner and Kozlov, 2021*; *Kiehn, 2016*; *Roberts et al., 2010*). These neurons are interconnected between members of the same 'module,' each of which is sequentially recruited as the animal adjusts its speed of locomotion. Neuromodulation tunes the recruitment of neurons within these modules in the adjustment of speed during locomotion (*Jha and Thirumalai, 2020*). In limbed animals, changes in speeds are often accompanied by changes in gait (*Bellardita and Kiehn, 2015*). How the corresponding qualitative and quantitative changes in the locomotor cycle are achieved is an area of active research. In the mouse spinal cord, V2a interneurons are required for maintaining left–right alternation at high-speed trotting (*Crone et al., 2009*). Furthermore, commissural $V0_V$ neurons are necessary for trot at all speeds, and ablation of commissural $V0_V$ and $V0_D$ neurons abolishes walk, trot, and gallop gaits (*Bellardita and Kiehn, 2015*). In the brainstem, the MLR controls the initiation of locomotion and the expression of specific gaits. The MLR's cuneiform nucleus (CnF) and the pedunculopontine nucleus mediate alternating locomotor stepping in mice, whereas the CnF alone is necessary for high-speed synchronous locomotion such as found in galloping (*Caggiano et al., 2018*). A recent study identified distinct subclasses of glutamatergic neurons within the MLR, each with distinct roles in motor control outside of locomotion (*Ferreira-Pinto et al., 2021*), suggesting that this nucleus has wider roles in regulating behavior. Despite this recent attention to the modulation of the speed of locomotion, the neural basis of the differential modulation of the locomotor cycle is still unknown. We have uncovered a set of inhibitory neurons, whose activity determines the duration of the interwave phase, thereby setting the frequency of locomotion. The inhibitory nature of this set of cells regulating muscle contractions has many parallels in other systems. One of the simplest circuit designs for rhythm generation, the 'half-center oscillator,' relies on reciprocal inhibition to generate alternating patterns of activity (*Marder and Bucher, 2001*), and reciprocal inhibition within the spinal cord is thought to underlie the generation of alternation during locomotion (*Deliagina and Orlovsky, 1980*; *Geertsen et al., 2011*; *Pratt and Jordan, 1987*). Indeed, inhibitory neurons shape the rhythms of neural activity on different timescales in systems from crustacean stomatogastric ganglion to vertebrate cortical circuits (*Cardin, 2019*; *Marder and Bucher, 2001*). In addition, a parallel between the *Drosophila* larval system and limbed locomotion can be seen in the mechanics of movement. For instance, in cats, the extensor muscles are mainly activated during the stance phase, while the flexor muscles are mainly activated in the swing phase (*Engberg and Lundberg, 1969*); similarly, we found that the fruit fly larva contracts its transverse muscles during the interwave phase, and its longitudinal muscles during the wave phase. These parallels may be mirrored within the neural circuitry mediating these muscle contractions. While the detailed implementation will obviously differ, the inhibitory neural circuit motif underlying the generation of the asymmetry of the two constituent phases of locomotion could therefore be conserved between species.

Our EM connectomics results also show that descending input has the potential to shape the activity of the speed modulation circuit. A31c receives strong synaptic input from descending neurons (e.g., S10 and T19x make up 32% of total input of A31cs in segments A2 and A3); the levels and timing of activity in A31c could therefore be modulated by descending cell types, which in turn could affect the activity of A26f and therefore the speed of crawling. It is also possible that this descending input acts to synchronize the activity of A31c across abdominal segments. Moreover, there are several cell types postsynaptic to A31c that send ascending projections from the VNC to thoracic and subesophageal segments (VNC-IN, A19f). These could be involved in relaying the state of the motor network,

for example, to provide higher brain centers with information on the current phase of the locomotor cycle or the intended speed of locomotion. Further exploration of the connectivity of A31c and its postsynaptic partner A26f will reveal how the activity of these descending and ascending projections relates to the speed modulation circuit.

# Materials and methods

## Key resources table

| Reagent type (species) or resource | Designation | Source or reference | Identifiers | Additional information |
|---|---|---|---|---|
| Genetic reagent (*Drosophila melanogaster*) | yw | Bloomington *Drosophila* Stock Center | BDSC #6598 | |
| Genetic reagent (*D. melanogaster*) | GMR24H08-GAL4.AD | Bloomington *Drosophila* Stock Center | BDSC #68300 | A31c-a8-sp.AD |
| Genetic reagent (*D. melanogaster*) | GMR45F08-GAL4.DBD | Bloomington *Drosophila* Stock Center | BDSC #70239 | A31c-a8-sp.DBD |
| Genetic reagent (*D. melanogaster*) | GMR44F09-GAL4.DBD | Bloomington *Drosophila* Stock Center | BDSC #71061 | A31c-sp.DBD |
| Genetic reagent (*D. melanogaster*) | GMR41F02-GAL4.AD | Bloomington *Drosophila* Stock Center | BDSC #75660 | A31c-sp.AD |
| Genetic reagent (*D. melanogaster*) | R76E09-LexA | Bloomington *Drosophila* Stock Center | BDSC #54951 | A26f-LexA |
| Genetic reagent (*D. melanogaster*) | VT050223-GAL4.AD | Bloomington *Drosophila* Stock Center | BDSC #72931 | A26f-sp.AD |
| Genetic reagent (*D. melanogaster*) | R15E05-GAL4.DBD | Bloomington *Drosophila* Stock Center | BDSC #68731 | A26f-sp.DBD |
| Genetic reagent (*D. melanogaster*) | GMR45F08-GAL4 | Bloomington *Drosophila* Stock Center | BDSC #49565 | A31c-a8-Gal4 |
| Genetic reagent (*D. melanogaster*) | GMR76E09-GAL4 | Bloomington *Drosophila* Stock Center | BDSC #39931 | A26f-GAL4 |
| Genetic reagent (*D. melanogaster*) | GMR41F02-LexA | Bloomington *Drosophila* Stock Center | BDSC #54794 | A31c-LexA |
| Genetic reagent (*D. melanogaster*) | nSyb-LexA_VK00027 | This study | | nSyb-LexA_VK27 |
| Genetic reagent (*D. melanogaster*) | eve[RRa-F]-GAL4 | Gift from Dr. Miki Fujioka | | |
| Genetic reagent (*D. melanogaster*) | sr-GAL4 | Bloomington *Drosophila* Stock Center | BDSC #26663 | |
| Genetic reagent (*D. melanogaster*) | UAS-CD4::GCaMP6f_attp40 | *Kohsaka et al., 2014* | | |
| Genetic reagent (*D. melanogaster*) | LexAop2-RGECO1_VK00005 | *Kohsaka et al., 2014* | | |
| Genetic reagent (*D. melanogaster*) | LexAop-jRCaMP1b | Bloomington *Drosophila* Stock Center | BDSC #64428 | |
| Genetic reagent (*D. melanogaster*) | 20XUAS-6XGFP | Bloomington *Drosophila* Stock Center | BDSC #52262 | |
| Genetic reagent (*D. melanogaster*) | *trans*-Tango | Bloomington *Drosophila* Stock Center | BDSC #77124 | |
| Genetic reagent (*D. melanogaster*) | MCFO-4 | Bloomington *Drosophila* Stock Center | BDSC #64088 | |
| Genetic reagent (*D. melanogaster*) | mhc-GFP | Gift from Dr. Cynthia L. Hughes | | |

*Continued on next page*

*Continued*

| Reagent type (species) or resource | Designation | Source or reference | Identifiers | Additional information |
|---|---|---|---|---|
| Genetic reagent (*D. melanogaster*) | UAS-CsChrimson::mVenus | Bloomington *Drosophila* Stock Center | BDSC #55136 | |
| Genetic reagent (*D. melanogaster*) | UAS-GtACR1_attp2 | Gift from Dr. Chris Doe | | |
| Genetic reagent (*D. melanogaster*) | UAS-VNC-CsChrimson | Gift from Dr. Karen Hibbard (*Hiramoto et al., 2021*) | | |
| Antibody | Anti-GFP (rabbit, polyclonal) | Frontier Institute | Af2020 | 1:1000 |
| Antibody | Anti-Fas2 (mouse, monoclonal) | Developmental Studies Hybridoma Bank | 1D4 | 1:10 |
| Antibody | Anti-GFP (guinea pig, polyclonal) | Frontier Institute | Af1180 | 1:1000 |
| Antibody | Anti-HA (rabbit, monoclonal) | Cell Signaling Technology | C29F4 | 1:1000 |
| Antibody | Anti-ChAT (mouse, monoclonal) | Developmental Studies Hybridoma Bank | 4B1 | 1:50 |
| Antibody | Anti-GABA (rabbit, polyclonal) | Sigma | A2052 | 1:100 |
| Antibody | Anti-VGluT (mouse, polyclonal) | Gift from Dr. Hermann Aberle | | 1:1000 |
| Antibody | Anti-DsRed (rabbit, polyclonal) | Clontech | #632496 | 1:500 |
| Antibody | Alexa Fluor 488-conjugated anti-rabbit IgG (goat, polyclonal) | Invitrogen Molecular Probes | A-11034 | 1:300 |
| Antibody | Alexa Fluor 555-conjugated anti-mouse IgG (goat, polyclonal) | Invitrogen Molecular Probes | A-21424 | 1:300 |
| Antibody | Alexa Fluor 488-conjugated anti-guinea pig IgG (goat, polyclonal) | Invitrogen Molecular Probes | A-11073 | 1:300 |
| Software, algorithm | FIJI | *Abràmoff et al., 2004* | RRID:SCR_002285 | |
| Software, algorithm | DeepLabCut | *Mathis et al., 2018* | https://github.com/DeepLabCut/DeepLabCut; (RRID:SCR_021391, version 2.1); *Mathis et al., 2023* | |

## Fly strains

Except where specifically mentioned, larvae were raised in standard cornmeal-based food at room temperature (25°C), and third-instar larvae were used for experiments. We used the following *all-trans* retinal (ATR) feeding conditions for optogenetics: 10 mM ATR yeast from 18 to 36 hr in CsChrimson and Channelrhodopsin 2 (Chr2.T159C) groups, 3 mM ATR yeast from 24 to 48 hr in GtACR1 groups. Fly strains are listed in Key resources table. We used the split GAL4 drivers *A31c-a8-sp* (*R24H08-GAL4.AD, R45F08-GAL4.DBD*), *A31c-sp* (*R41F02-GAL4.AD, R44F09-GAL4.DBD*), and *A26f-sp* (*VT050223-GAL4.AD, R15E05-GAL4.DBD*). Transgenic flies *nSyb-LexA* were generated in the lab. The enhancer sequence of *neuronal Synaptobrevin (nSyb)* (*R57C10*, *Pfeiffer et al., 2012*) was cloned into pBPLexA::p65Uw plasmid (*Pfeiffer et al., 2010*). The transgenic line was generated in the *VK00027* locus (BestGene Inc, USA). Sources of the fly strains are listed in Key resources table.

## Immunostaining and calcium imaging

We used a standard immunostaining procedure (*Kohsaka et al., 2014*). First, the larvae were dissected in the fillet preparation, fixed in 4% formaldehyde for 30 min at room temperature, washed twice with 0.2% Triton X-100 in PBS (PBT) for 15 min at room temperature, blocked with 5% normal goat serum

in PBT for 30 min at room temperature, and stained with the first antibody at 4°C for 24–48 hr. Then, the preparations were washed twice with PBT for 15 min and stained with the second antibody at 4°C for 24–48 hr. Sources and concentrations of antibodies are listed in Key resources table.

In the calcium imaging of the isolated CNS, the CNS of third-instar larvae was dissected out (*Kohsaka et al., 2014*), transferred to a drop of TES buffer (TES 5 mM, NaCl 135 mM, KCl 5 mM, MgCl₂ 4 mM, CaCl₂ 2 mM, sucrose 36 mM; pH = 7.15), and attached dorsal-up on MAS-coated slide glass for imaging (Matsunami Glass, Japan). GCaMP6f fluorescence was detected by a spinning-disk confocal unit (CSU21, Yokogawa, Japan) and an EMCCD camera (iXon, Andor Technology, Germany) on an upright microscope, Axioskop2 FS (Zeiss, Germany). We used a dual-view system (CSU-DV, Solution Systems, Japan) to perform dual-color calcium imaging for GCaMP and R-GECO1.

## Top-view crawling assay and analysis

Third-instar wandering larvae of *sr-GAL4>20xUAS-6xGFP* about 0–4 hr after the start of wandering were used. We transferred a larva onto an agarose plate of a standard concentration (1.5%), waited for about 1 min, and took a video for 5 min. An Olympus stereomicroscope (SZX16, Olympus, Japan) and a ×0.7 lens were used for magnification. A CMOS camera (C11440-22CU, Hamamatsu Photonics, Japan) was used for video recording. A square of 1.6 × 1.6 cm of 1024 × 1024 pixels was recorded. The frame rate was set at 30 Hz. A mercury lamp (U-HGLGPS, Olympus) and an excitation filter (460–495 nm) were used to deliver ~5 µW/mm² of blue light for illumination.

We reviewed all videos to extract episodes of straight runs of more than three strides. We then randomly selected three episodes for each larva and analyzed the stride parameters. An ImageJ script was used to manually annotate the video to obtain kinematic parameters (version 1.53, *Abràmoff et al., 2004*). The stride length was obtained from the distance between the landing positions of the prominent ventral denticle at A8 on one lateral side. The stride duration was obtained from the duration between the unhooking moments. The time of wave initiation was annotated when the A8 prominent denticle moved half a segmental length. The speed was calculated by dividing the total stride lengths by the total stride durations in an episode.

To model the relationship between the stride duration and the duration of the two constituent phases, we tested the polynomial models and the piecewise linear model with two pieces. We then compared the Bayesian information criterion (BIC) between these models (*Burnham and Anderson, 2004*). The BIC is defined as

$$BIC = K \ln(n) - 2 \ln(\hat{L}).$$

where $K$ is the number of estimated parameters in the model, $n$ is the amount of data, $\hat{L}$ is the maximum value of the likelihood function for the model. In the case of least-squares estimation with normally distributed errors, BIC can be expressed as

$$BIC = K \ln(n) + n \ln(\hat{\sigma}^2),$$

where $\hat{\sigma}^2$ is the average of the squares of residuals. We calculated the BIC for the linear piecewise model of two pieces and the polynomial models of degrees from 2 to 10. The BIC has a minimum value with the cubic polynomial model.

## Side-view imaging of the muscular ends and analysis

Third-instar wandering larvae about 0–12 hr after starting wandering were used. An agarose plate of a standard concentration (1.5%) with black ink (0.2%) was used as the substrate. We oriented a CMOS camera (C11440-22CU, Hamamatsu, Japan) and its zoom lens (MLM3X-MP, Computar, Japan) with a 2× extender (FP-EX2, RICOH, Japan) horizontally for recording. Each time one larva was transferred to the agarose plate for recording. We manually moved the plate to let the camera focus on the larval body wall. The top-view imaging was simultaneously recorded with the same instrument described in the previous method section. A mercury lamp (U-HGLGPS, Olympus) and an excitation filter (460–495 nm) were used to deliver 5 µW/mm² of blue light for the illumination of the GFP-tagged tendon cells. We recorded at 30 Hz for about 3 min and typically collected 3–5 episodes in focus. Each episode includes 2–5 straight crawls.

We reviewed all videos to select two episodes for each larva that had the best focus and analyzed the stride parameters. To visualize the kinematics of the muscular movement, we used DeepLabCut (*Mathis et al., 2018*) to track the muscular ends in muscles LT2, VL4, and DO1. We labeled the muscular ends for 40–50 frames in each video and trained the resnet50 network with the labeled frames for 1,000,000 iterations. To understand the relationship between the contraction of LT muscles and the head and tail movement, an ImageJ script was used to obtain the minimum/maximum length of the LT2 muscle, the stride length, and the interwave duration (*Abràmoff et al., 2004*). To obtain the minimum/maximum length of the LT2 muscle, we annotated the position of the muscular ends of the LT2 muscle in segments A2-A7 when they were mostly contracted and extended and calculated the distance of the pairs of muscular ends. To obtain the stride length, we annotated the landing positions of the tail and calculated the distance. The interwave duration was obtained as described in the previous section. The speed was calculated by dividing the stride length by the stride duration.

## Trans-synaptic tracing by *trans*-Tango

As *trans*-Tango expression is leaky in larval ventral nerve cord (VNC) neurons when using the recommended rearing temperature 18°C (*Talay et al., 2017*), *trans*-Tango larvae were incubated at 30°C for 1 d before the experiment. *trans*-Tango expression was thereby restricted to a small number of neurons in combination with the split GAL4 driver *A31c-a8*. We then identified each single neuron by comparing its morphology to the EM database (*Ohyama et al., 2015*).

## EM reconstruction

Serial sectioning transmission electron microscopy (ssTEM) data were analyzed as described in *Ohyama et al., 2015*. Briefly, reconstructions were made in a modified version of CATMAID (*Saalfeld et al., 2009*; http://www.catmaid.org). LT motoneurons and their presynaptic partners had been identified and reconstructed previously within the ssTEM volume (*Zwart et al., 2016*). These reconstructions were used to identify and reconstruct all presynaptic partners.

## Measurement and quantification of calcium activity

To analyze calcium imaging data, we manually circled regions of interest (ROIs) using ImageJ (version 1.53, *Abràmoff et al., 2004*). ROIs were chosen at the medial dendritic sites for the A26f neurons, at the axons for the A31c neurons, and the neuropil for the pan-neuronal line in each neuromere. To compare the calcium imaging of different forward cycles, we normalized the time in *Figures 2D and 4D* relative to the peak ΔF/F of nSyb in segments A4 and A1. We normalized the time in *Figure 4F′* relative to the peak ΔF/F of A31c in A4 preceding the FW and the peak ΔF/F of A31c in A4 during the FW. To obtain the time-lagged cross-correlation, we slide a trace of calcium activity as in *Figure 2D* or *Figure 4D*, calculated the Pearson correlation coefficients with traces of calcium activity in other segments, and calculated the mean value of correlation coefficients by using Fisher-z correction.

## Optogenetic assay of free crawling

We assayed the response of larvae to optogenetic stimulation by using the same imaging system as the top-view imaging assay. The background illumination and the light for the optogenetic stimulation were set as the following. In the GtACR1 groups, we used a 590 nm LED of ~150 μW/mm$^2$ to provide the optogenetic stimulation, while a 660 nm LED (M660L3, Thorlabs, USA) or an infrared light (LDQ-150IR2-850, CCS, Japan) provided the background illumination. In the CsChrimson groups, we used an 850 nm infrared light (LDQ-150IR2-850, CCS) of ~40 μW/mm$^2$ to provide the background light and used the 660 nm LED to apply the optogenetic stimulation of ~60 μW/mm$^2$. We used an ImageJ script to manually annotate videos to obtain the kinematic parameters (version 1.53, *Abràmoff et al., 2004*). In the experiments using GtACR1, the larva can show transient turning or stopping responses to 590 nm light. In these groups, we analyzed strides if forward cycles were not halted or after forward cycles were reinitiated. In the experiment using *A26f-sp* drivers, we only analyzed the data when the *GtACR1/CsChrimson* was expressed in more than four A26f neurons, which was determined by post hoc staining.

## Assay of muscular response to optogenetic stimulation in the fillet and sideways preparation

We used a semi-intact fillet preparation to assay muscular responses to optogenetic activation (*Kohsaka et al., 2014*). After the preparation, we waited for about 10 min, until the larva stopped its frequent spontaneous axial waves.

To constrain the movement of the larva without impairing peristaltic behavior and visualize the lateral side of the larva, we devised a new preparation named sideways preparation. In this preparation, the larva is fixed by two pins on a vertical side of a polydimethylsiloxane (PDMS; Silpot 184, Toray, Japan) plate and oriented lateral side up to visualize the LT muscles. The larva can show spontaneous forward peristalsis-like behavior in this preparation. In the preparation, we prepared a PDMS plate with a standing PDMS island filled with 4°C TES buffer, transferred a larva to the PDMS plate, and used two pins to fix the head and tail of the third-instar larvae (*Figure 5A and B*). The tail was pinned to the bottom PDMS substrate to make the pin perpendicular to the larval sagittal plane with two pricking points close to the two prominent lateral denticles in the A8 segment. The head was pinned to the PDMS island to make the pin perpendicular to the larval frontal plane. After pinning, the PDMS island was attached to the tail pin and supported the ventral larval body. 4°C TES buffer was used to reduce the larval motion during the preparation. We changed the buffer to 25°C before imaging.

A local stimulation microscope was used for muscular imaging and optogenetic stimulation (*Matsunaga et al., 2013*; *Takagi et al., 2017*). The microscope (FV1000, Olympus) has two separate optical paths for muscular imaging and optical stimulation, respectively: blue light from a Xeon lamp (X-Cite exacte, Excelitas Technologies, USA) and a GFP dichroic mirror (U-MGFP/XL, Olympus), which were used to image the muscles in the abdominal segments A3/A4 to A7/A8, and a scanning laser of blue (488 nm) or green (559 nm) light, which was used to stimulate the CNS optogenetically. A dichroic mirror separates the two optical paths. To fit the larva into the field of view, we used a 4x Olympus objective and a 1× or a 0.63× adapter. Muscular contractions were recorded by an EMCCD camera (iXon, Andor Technology). We used different combinations of optogenetic stimulation and muscular illumination. In the sideways preparation, a rectangular scanning of about 0.85 mm × 0.4 mm by the 559 nm laser was used for optogenetic stimulation (~20 µW/mm$^2$ for the CsChrimson groups and ~40 µW/mm$^2$ for the GtACR1 groups), while blue light of ~10 µW/mm$^2$ was used for muscular illumination. In the fillet preparation, a 'tornado' scanning of a radius of ~0.3 mm by a 488 nm laser was used to activate the Chr2 (~40 mW/mm$^2$), while a blue light of ~50 µW/mm$^2$ was used for muscular illumination. DeepLabCut (*Mathis et al., 2018*) was used to track the muscular ends. We labeled the muscular ends in 40–50 frames in each video and trained them using the resnet50 network. The neural network was trained 1,000,000 times.

## Statistical tests for the optogenetic experiments

We estimated the sample sizes according to conventions in this field. Changes in most values (stride duration, stride length, etc.) were directly used for statistical tests except that the changes in muscle length were normalized to the minimal length within each animal. We tested the significance of the changes before and after the optogenetic manipulation and compared the changes between the experimental and the control. As each animal was treated with optogenetic stimulation multiple times, to increase the statistical power and avoid Type I error (false positive), we used hierarchical bootstrapping methods for the comparison before and after the optogenetic stimulation and the comparison between the experimental and control group (*Saravanan et al., 2020*). To generate the bootstrapped dataset, we resampled data from the experimental dataset 10,000 times. Each time we (1) resample $n$ animals with replacement ($n$ is the animal number in the experiment), and (2) resample $m_1, .., m_n$ trials within animals with replacement ($m_i$ is the number of trials of the resampled animal $i$ in the experiment) (*Saravanan et al., 2020*). For the comparison before and after the optogenetic stimulation, we used the empirical method by (1) computing the animal-wide mean of the bootstrapped sample $\mu^*$, (2) computing the difference between $\mu^*$ and the animal-wide mean of the experiment µ, and (3) computing the p-value as the quantile of µ in $\mu^* - \mu$ *Efron and Tibshirani, 1994*. For the comparison between the experimental and control group, we (1) computed the animal-wide mean of the two bootstrapped samples ($\mu^{a,*}$ and $\mu^{b,*}$), (2) computed a joint probability distribution of $\mu^{a,*}$ and $\mu^{b,*}$, and (3) computed the p-value as the density of the joint probability (*Saravanan et al., 2020*). All

analysis was done with Python (version 3.9.12) scripts using the libraries NumPy (version 1.21.5) and SciPy (version 1.7.3). Asterisks represent the range of p-values ($*p<0.05$; $**p<0.005$; $***p<0.0005$).

## Acknowledgements

We thank Bloomington *Drosophila* Stock Center, KYOTO *Drosophila* Stock Center, and Drs. Chris Doe, Miki Fujioka, Karen Hibbard, and Cynthia L Hughes for the fly lines (Key resources table). We thank Developmental Studies Hybridoma Bank and Dr. Hermann Aberle for the antibodies. We thank Drs. Stefan Pulver and Wenchang Li for critical comments on the paper. We thank Mr. Takahisa Date for supporting immunostaining. Finally, we give our heartfelt thanks to Dr. Albert Cardona for continued access to the L1 EM dataset. This work was supported by MEXT/JSPS KAKENHI grants (17K19439, 19H04742, 20H05048 to AN and 17K07042, 20K06908 to HK) and the Royal Society of Edinburgh (grant 64553 to MFZ).

## Additional information

### Funding

| Funder | Grant reference number | Author |
|---|---|---|
| Japan Society for the Promotion of Science | KAKENHI 17K19439 | Akinao Nose |
| Japan Society for the Promotion of Science | KAKENHI 17K07042 | Hiroshi Kohsaka |
| Royal Society of Edinburgh | grant 64553 | Maarten F Zwart |
| Japan Society for the Promotion of Science | KAKENHI 19H04742 | Akinao Nose |
| Japan Society for the Promotion of Science | KAKENHI 20H05048 | Akinao Nose |
| Japan Society for the Promotion of Science | KAKENHI 20K06908 | Hiroshi Kohsaka |

The funders had no role in study design, data collection and interpretation, or the decision to submit the work for publication.

### Author contributions

Yingtao Liu, Conceptualization, Data curation, Software, Formal analysis, Investigation, Visualization, Methodology, Writing - original draft, Writing - review and editing; Eri Hasegawa, Formal analysis; Akinao Nose, Supervision, Funding acquisition; Maarten F Zwart, Conceptualization, Data curation, Formal analysis, Supervision, Funding acquisition, Validation, Investigation, Visualization, Methodology, Writing - original draft, Writing - review and editing; Hiroshi Kohsaka, Conceptualization, Resources, Formal analysis, Supervision, Funding acquisition, Validation, Investigation, Visualization, Methodology, Writing - original draft, Project administration, Writing - review and editing

### Author ORCIDs

Yingtao Liu  http://orcid.org/0000-0002-5090-8784
Maarten F Zwart  http://orcid.org/0000-0002-5073-8631
Hiroshi Kohsaka  http://orcid.org/0000-0003-1087-9680

### Decision letter and Author response

Decision letter https://doi.org/10.7554/eLife.83328.sa1
Author response https://doi.org/10.7554/eLife.83328.sa2

## Additional files

### Supplementary files
• MDAR checklist

### Data availability
The data generated in this study are available from the Zenodo repository (https://doi.org/10.5281/zenodo.7052205).

The following dataset was generated:

| Author(s) | Year | Dataset title | Dataset URL | Database and Identifier |
|---|---|---|---|---|
| Liu Y, Hasegawa E, Nose A, Zwart M, Kohsaka H | 2022 | Dataset for "Synchronous multi-segmental activity between metachronal waves controls locomotion speed in *Drosophila* larvae" | https://zenodo.org/record/8067638 | Zenodo, 10.5281/zenodo.7052205 |

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
