## [Editor Report]

Exploiting the power of the *Drosophila* larva as a model, this fundamental study sheds light on the neuronal mechanisms of speed regulation during locomotion. The data obtained using a combination of functional and structural approaches are rigorous and convincing. The identified mechanism of speed regulation could be shared with limbed animals and therefore this work is of relevance to those studying the motor system and locomotion across species.

---

## [Decision Letter]

**Decision letter after peer review:**

Thank you for submitting your article "Synchronous multi-segmental activity between metachronal waves controls locomotion speed in *Drosophila* larvae" for consideration by *eLife*. Your article has been reviewed by 3 peer reviewers, and the evaluation has been overseen by a Reviewing Editor and K VijayRaghavan as the Senior Editor. The following individual involved in the review of your submission has agreed to reveal their identity: Jimena Berni (Reviewer #3).

Essential revisions:

Liu et al. investigated the mechanism of speed control in larval fruit flies in which movement is generated by the coordination of muscle contraction across several segments forming a peristaltic wave. The authors showed very convincing behavioral data demonstrating that the inter-wave phase is regulated to control speed. They found that one of the Lateral transverse muscles (LT2) is constantly contracted during the inter-wave phase. The authors further describe two inhibitory neurons projecting onto LT2 motoneuron (A31c and A26f) that exhibit patterns of activity suggesting that they could be involved in the process. Overall, it is a very interesting paper that describes a new mechanism of speed control, but several points need to be solved before the manuscript is ready for publication:

1) When the neuronal activity is manipulated affecting the contraction of the motoneuron LT2, the interwave time and the speed of locomotion are altered. Data describing the pattern of activity of A31c and A26f neurons in the isolated nervous system is not completely convincing, due to the clear overlap of the neuronal activity with the contraction of abdominal segments.

2) The number of some of the behavioural experiments is very low. Add more.

3) The authors show the A26f and A31c neurons have an opposite role in crawling and LT muscle contractions and the existing EM connectomics data. Experiments of dual color imaging showed that the activity of A26f is bookended to peaks of A31c activity in adjacent segments, however, the authors to not causatively show that A31c inhibits A26f. The authors could for instance activate A31C and image in A26f. I do not impose to do those experiments, especially in case genetic lines to do them do not already exist or would be difficult to make in a reasonable time. However, if such experiments cannot be performed, the author should tone down the direct dependence of inhibition of A26f on A31c (alone), as the authors do not directly show that the inhibition of A26f by A31c is responsible for the changes in LT contractions.

4) Some findings from EM connectomics could be discussed further. For example, A31c receives synaptic inputs from descending neurons (E.g. S10). This could suggest that A31c could modulate the speed of locomotion depending on these inputs, which could inhibit A31c to promote faster crawling or promote its activity for slower crawling. Likewise, what would be the putative role of the ascending neurons? How do A31c from different segments synchronize their activity: do they synapse onto A31c of adjacent segments? It could be beneficial to discuss and speculate on some of the information gained from connectivity analysis further.

5) The text lacks clarity and should be improved:

a) The authors should clarify the expression patterns of the lines that they used in each of the activation/silencing experiments. They show images on VNC in these lines. Do these lines label selectively the relevant neurons or where there any off-targets in the SEZ/brain, as those could have an effect on the phenotypes? Since they use UAS-VNC Chrimson for A19c optogenetic activation in Figure 7 supplement, it suggests that this line has off-targets in the SEZ and/or brain. Given the EM connectomics and functional imaging data to complement the optogenetic manipulation data, any off-targets would not necessarily change the interpretation of the experiments or the conclusion but this caveat if it exists should be mentioned (e.g. optogenetic inactivation of A31c?).

b) Line 130-140 the authors suggest that when the stride duration is approximately 1 s the inter-wave duration is minimal and the wave phase reduces with stride duration and when it is above 1.2 s the wave duration is constant and the inter-wave duration increases with stride duration.Given that these thresholds are approximate and they might depend on other factors, like the size of the larva, etc. could the authors formulate this differently? Maybe something along the lines of for example saying that the stride duration decreases as the inter-wave duration decrease while the wave duration remains constant and when the inter-wave duration becomes minimal, then the stride duration can further decrease depending on the duration of the wave that is no longer constant.

c) L310-312 authors say that there is a synchronous activity in neurons A26f from neuromeres a3-a5. However, the authors couldn't examine the activity in all segments as the line labels only A26f in these segments (a3-a5). The way it is written it is not clear that they only observe a3-a5 synchronous activity because they genetically restrict GCAMP to these cells. This should be explained more clearly in the text.

6) The authors found that A26f and A31c are GABA-ergic. Did they check for other neurotransmitters and they found both of these neurons are negative or did they only check for GABA?

Note: Consider abstracting the connectivity motif to a GABA-ergic neuron inhibiting a GABA-ergic pre-motor neuron instead of A31c inhibiting A26f and add that the pre-pre-motor GABAergic neuron receives descending input.

*Reviewer #1 (Recommendations for the authors):*

1) Experiments of dual color imaging showed that the activity of A26f is bookended to peaks of A31c activity in adjacent segments, however, the authors do not causatively show that A31c inhibits A26f, by for instance activating A31C and imaging in A26f. Given that the authors show these neurons have an opposite role on crawling and LT muscle contractions and the existing EM connectomics data, I am not necessarily suggesting doing those experiments, in case genetic lines to do them do not already exist or would be difficult to make in a reasonable time. However, if such experiments cannot be performed, the author should tone down the direct dependence of inhibition of A26f on A31c (alone), as the authors do not directly show that the inhibition of A26f by A31c is responsible for the changes in LT contractions.

2) Some findings from EM connectomics could be discussed further. For example, A31c receives synaptic inputs from descending neurons (E.g. S10). This could suggest that A31c could modulate the speed of locomotion depending on these inputs, which could inhibit A31c to promote faster crawling or promote its activity for slower crawling. Likewise, what would be the putative role of the ascending neurons? How do A31c from different segments synchronize their activity: do they synapse onto A31c of adjacent segments? It could be beneficial to discuss and speculate on some of the information gained from connectivity analysis further.

3) The authors should clarify the expression patterns of the lines that they used in each of the activation/silencing experiments. They show images on VNC in these lines. Do these lines label selectively the relevant neurons or where there any off-targets in the SEZ/brain, as those could have an effect on the phenotypes? Since they use UAS-VNC Chrimson for A19c optogenetic activation in Figure 7 supplement, it suggests that this line has off-targets in the SEZ and/or brain. Given the EM connectomics and functional imaging data to complement the optogenetic manipulation data, any off-targets would not necessarily change the interpretation of the experiments or the conclusion but this caveat if it exists should be mentioned (e.g. optogenetic inactivation of A31c?).

4) Line 130-140 the authors suggest that when the stride duration is approximately 1 s the inter-wave duration is minimal and the wave phase reduces with stride duration and when it is above 1.2 s the wave duration is constant and the inter-wave duration increases with stride duration.

Given that these thresholds are approximate and they might depend on other factors, like the size of the larva, etc. could the authors formulate this differently? Maybe something along the lines of for example saying that the stride duration decreases as the inter-wave duration decrease while the wave duration remains constant and when the inter-wave duration becomes minimal, then the stride duration can further decrease depending on the duration of the wave that is no longer constant.

5) The authors found that A26f and A31c are GABA-ergic. Did they check for other neurotransmitters and they found both of these neurons are negative or did they only check for GABA?

6) L310-312 authors say that there is a synchronous activity in neurons A26f from neuromeres a3-a5. However, the authors couldn't examine the activity in all segments as the line labels only A26f in these segments (a3-a5). The way it is written it is not clear that they only observe a3-a5 synchronous activity because they genetically restrict GCAMP to these cells. This should be explained more clearly in the text.

7) I would suggest abstracting this connectivity motif to GABA-ergic neuron inhibit a GABA-ergic pre-motor neuron instead of A31c inhibit A26f and add that the pre-pre-motor GAB-ergic neuron receives descending input.

---

## [Author Response]

Essential revisions:Liu et al. investigated the mechanism of speed control in larval fruit flies in which movement is generated by the coordination of muscle contraction across several segments forming a peristaltic wave. The authors showed very convincing behavioral data demonstrating that the inter-wave phase is regulated to control speed. They found that one of the Lateral transverse muscles (LT2) is constantly contracted during the inter-wave phase. The authors further describe two inhibitory neurons projecting onto LT2 motoneuron (A31c and A26f) that exhibit patterns of activity suggesting that they could be involved in the process. Overall, it is a very interesting paper that describes a new mechanism of speed control, but several points need to be solved before the manuscript is ready for publication:1) When the neuronal activity is manipulated affecting the contraction of the motoneuron LT2, the interwave time and the speed of locomotion are altered. Data describing the pattern of activity of A31c and A26f neurons in the isolated nervous system is not completely convincing, due to the clear overlap of the neuronal activity with the contraction of abdominal segments.

We appreciate that this needs to be addressed. There are at least two things to consider:

First, we think we have not adequately discussed the two phases of the locomotor cycle, which we think is the cause of confusion that has led to our data being in apparent conflict. Supplementary figure 1 describes the contraction patterns of LT2, VL4, and DO1 in segments A2-A7. Here you can see that, during the interwave phase, in the posterior-most segments (A6-7 in this case) there are contractions not only in LT2, but also in DO1, which is a longitudinal muscle. We believe this occurs in other longitudinal muscles in posterior segments, as well. This is different from what happens in the more anterior segments, where contractions are restricted to the LTs during the interwave phase. We think these posterior longitudinal contractions (which we don’t explore in this paper) might have a different function to the contractions in the more anterior segments, where they drive the forwards peristaltic movements of each segment. We would argue that the synchronous peaks in activity in the *nsyb* expression pattern, which we use as a readout of fictive behaviour, is caused by the activity in cells that drive the contractions in those muscles, and therefore occur during the interwave phase. This explains why the activity of both A26f and A31c coincides with the peaks in activity in the *nsyb* pattern in the posterior segments. We have made modifications in the text to address this issue.

We have included a greater discussion of the muscle contraction patterns that are depicted in Supplementary figure 1 (lines 152-163), adjusted the wording of our description of the activity of A26f and A31c (lines 203-216 and 265-277), and included a discussion of our interpretation of the *nsyb* activity in the posterior segments in relation to the observed muscle contraction patterns and phases of the locomotor cycle (lines 428-437).

Second, the two types of experiments rely on different preparations, one on fully intact animals (the manipulation of neural activity and characterisation of the behavioral consequences), the other ex vivo CNS preparations (activity imaging). There will be some differences between the two, including the presence (or absence) of sensory feedback. This could mean that the neural activity in the imaging experiments does not entirely follow its normal patterns. This potential difference cannot be avoided; however, we believe that given the wide-spread use and validation of the imaging technique, the consistent results between the different techniques in our paper (including muscle imaging, optogenetic manipulations, and connectomics), and the above point on muscle contractions in the posterior segments, that our interpretation of our results is reasonable.

2) The number of some of the behavioural experiments is very low. Add more.

We agree that the sample size is on the low end for some of our experiments. We have added to our datasets so that now each experiment has the following sample sizes:

The activation of A26f in the sideways preparation (Figure 5D) the experimental group (n = 8 larvae), the control group (n = 8 larvae)The activation of A31c in the fillet preparation (Figure 5H) the experimental group (n = 8 larvae), the control group (n = 8 larvae)The inhibition of A26f/A31c in the sideways preparation (Figure 6) A26f group (n = 8 larvae), A31c group (n=8 larvae), the control group (n = 8 larvae)The activation of A26f in free crawling larvae (Figure 7C) the experimental group (n = 10 larvae), the control group (n = 10 larvae)The inhibition of A26f in free crawling larvae (Figure 7D) the experimental group (n = 10 larvae), the control group (n = 10 larvae)The activation of A31c in free crawling larvae (Figure 7E) the experimental group (n = 10 larvae), the control group (n = 10 larvae)The inhibition of A31c in free crawling larvae (Figure 7F) the experimental group (n = 10 larvae), the control group (n = 10 larvae)

3) The authors show the A26f and A31c neurons have an opposite role in crawling and LT muscle contractions and the existing EM connectomics data. Experiments of dual color imaging showed that the activity of A26f is bookended to peaks of A31c activity in adjacent segments, however, the authors to not causatively show that A31c inhibits A26f. The authors could for instance activate A31C and image in A26f. I do not impose to do those experiments, especially in case genetic lines to do them do not already exist or would be difficult to make in a reasonable time. However, if such experiments cannot be performed, the author should tone down the direct dependence of inhibition of A26f on A31c (alone), as the authors do not directly show that the inhibition of A26f by A31c is responsible for the changes in LT contractions.

This is a very good comment, and we agree that we have no direct evidence that A31c inhibits A26f. This means that, for instance, our observation that optogenetically activating A31c induces relaxation of transverse muscles could be explained by A31c’s potential inhibition of its other postsynaptic partners. The genetics of doing the experiments to provide direct evidence for A31c’s inhibition of A26f are quite complex and time-consuming. We appreciate the reviewer’s suggestion of toning down the wording in our manuscript. We have therefore adjusted our phrasing to temper our conclusions (as in lines 367-379).

4) Some findings from EM connectomics could be discussed further. For example, A31c receives synaptic inputs from descending neurons (E.g. S10). This could suggest that A31c could modulate the speed of locomotion depending on these inputs, which could inhibit A31c to promote faster crawling or promote its activity for slower crawling. Likewise, what would be the putative role of the ascending neurons? How do A31c from different segments synchronize their activity: do they synapse onto A31c of adjacent segments? It could be beneficial to discuss and speculate on some of the information gained from connectivity analysis further.

We agree that this discussion adds significantly to the paper. We have added a section on this within the discussion (lines 488-500).

5) The text lacks clarity and should be improved:a) The authors should clarify the expression patterns of the lines that they used in each of the activation/silencing experiments. They show images on VNC in these lines. Do these lines label selectively the relevant neurons or where there any off-targets in the SEZ/brain, as those could have an effect on the phenotypes? Since they use UAS-VNC Chrimson for A19c optogenetic activation in Figure 7 supplement, it suggests that this line has off-targets in the SEZ and/or brain. Given the EM connectomics and functional imaging data to complement the optogenetic manipulation data, any off-targets would not necessarily change the interpretation of the experiments or the conclusion but this caveat if it exists should be mentioned (e.g. optogenetic inactivation of A31c?).

We agree that the expression in the SEG/brain should be demonstrated. In the revised manuscript, the expression of A26f-sp is shown in Figure 7 supplement 2A, and that of A31csp is shown in Figure 7 supplement 2C and 2D. A26f-sp expression is sporadic in the brain and is not found in the SEG. A31c-sp expression is limited to a few cell clusters in the brain and sporadic cells in the SEG. The A31c-sp brain cells are predominantly immature and nonfunctional neurons. For the gain of function experiments, we suppressed the expression in the SEG/brain by using UAS-VNC-Chrimson which drives the gene expression only within the ventral nerve cord or applying optogenetic stimulus confined within the ventral nerve cord. In the loss of function experiment, where there was no obvious phenotype (Figure 7F7F’’), the expression in the SEG/brain might affect the results. This observation is described in lines 372-377.

b) Line 130-140 the authors suggest that when the stride duration is approximately 1 s the inter-wave duration is minimal and the wave phase reduces with stride duration and when it is above 1.2 s the wave duration is constant and the inter-wave duration increases with stride duration.Given that these thresholds are approximate and they might depend on other factors, like the size of the larva, etc. could the authors formulate this differently? Maybe something along the lines of for example saying that the stride duration decreases as the inter-wave duration decrease while the wave duration remains constant and when the inter-wave duration becomes minimal, then the stride duration can further decrease depending on the duration of the wave that is no longer constant.

We agree that this should be rephrased. We have therefore made changes to the text according to the reviewer’s suggestion, which we think address this point adequately (now lines 136-142).

c) L310-312 authors say that there is a synchronous activity in neurons A26f from neuromeres a3-a5. However, the authors couldn't examine the activity in all segments as the line labels only A26f in these segments (a3-a5). The way it is written it is not clear that they only observe a3-a5 synchronous activity because they genetically restrict GCAMP to these cells. This should be explained more clearly in the text.

We agree that this is not clear. We have amended the text to make this more apparent (lines 265-268).

6) The authors found that A26f and A31c are GABA-ergic. Did they check for other neurotransmitters and they found both of these neurons are negative or did they only check for GABA?

Indeed, we performed immunohistochemistry for GABA, vGlut, and ChAT (see figures 2 and 4 – supplement). These two cell types are positive for GABA, but negative for the other two markers; together these three represent the main neurotransmitters in the *Drosophila* larval VNC.

Note: Consider abstracting the connectivity motif to a GABA-ergic neuron inhibiting a GABA-ergic pre-motor neuron instead of A31c inhibiting A26f and add that the pre-pre-motor GABAergic neuron receives descending input.

We agree with this excellent suggestion. We have modified the connectivity diagram in figure 8 to reflect this.

Reviewer #1 (Recommendations for the authors):1) Experiments of dual color imaging showed that the activity of A26f is bookended to peaks of A31c activity in adjacent segments, however, the authors do not causatively show that A31c inhibits A26f, by for instance activating A31C and imaging in A26f. Given that the authors show these neurons have an opposite role on crawling and LT muscle contractions and the existing EM connectomics data, I am not necessarily suggesting doing those experiments, in case genetic lines to do them do not already exist or would be difficult to make in a reasonable time. However, if such experiments cannot be performed, the author should tone down the direct dependence of inhibition of A26f on A31c (alone), as the authors do not directly show that the inhibition of A26f by A31c is responsible for the changes in LT contractions.

We thank the reviewer for this suggestion. We have revised the manuscript as suggested. Please see the response to Essential revision 3 described above.

2) Some findings from EM connectomics could be discussed further. For example, A31c receives synaptic inputs from descending neurons (E.g. S10). This could suggest that A31c could modulate the speed of locomotion depending on these inputs, which could inhibit A31c to promote faster crawling or promote its activity for slower crawling. Likewise, what would be the putative role of the ascending neurons? How do A31c from different segments synchronize their activity: do they synapse onto A31c of adjacent segments? It could be beneficial to discuss and speculate on some of the information gained from connectivity analysis further.

We thank the reviewer for raising this point. We revised the manuscript as suggested. Please see the response to Essential revision 4 described above.

3) The authors should clarify the expression patterns of the lines that they used in each of the activation/silencing experiments. They show images on VNC in these lines. Do these lines label selectively the relevant neurons or where there any off-targets in the SEZ/brain, as those could have an effect on the phenotypes? Since they use UAS-VNC Chrimson for A19c optogenetic activation in Figure 7 supplement, it suggests that this line has off-targets in the SEZ and/or brain. Given the EM connectomics and functional imaging data to complement the optogenetic manipulation data, any off-targets would not necessarily change the interpretation of the experiments or the conclusion but this caveat if it exists should be mentioned (e.g. optogenetic inactivation of A31c?).

We thank the reviewer for pointing out this issue. We revised the manuscript as suggested. Please see the response to Essential revision 5a described above.

4) Line 130-140 the authors suggest that when the stride duration is approximately 1 s the inter-wave duration is minimal and the wave phase reduces with stride duration and when it is above 1.2 s the wave duration is constant and the inter-wave duration increases with stride duration.Given that these thresholds are approximate and they might depend on other factors, like the size of the larva, etc. could the authors formulate this differently? Maybe something along the lines of for example saying that the stride duration decreases as the inter-wave duration decrease while the wave duration remains constant and when the inter-wave duration becomes minimal, then the stride duration can further decrease depending on the duration of the wave that is no longer constant.

We thank the reviewer for this suggestion. We revised the manuscript as suggested. Please see the response to Essential revision 5b described above.

5) The authors found that A26f and A31c are GABA-ergic. Did they check for other neurotransmitters and they found both of these neurons are negative or did they only check for GABA?

We thank the reviewer for raising this point. We revised the manuscript as suggested. Please see the response to Essential revision 6 described above.

6) L310-312 authors say that there is a synchronous activity in neurons A26f from neuromeres a3-a5. However, the authors couldn't examine the activity in all segments as the line labels only A26f in these segments (a3-a5). The way it is written it is not clear that they only observe a3-a5 synchronous activity because they genetically restrict GCAMP to these cells. This should be explained more clearly in the text.

We thank the reviewer for pointing out this issue. We revised the manuscript as suggested. Please see the response to Essential revision 5c described above.

7) I would suggest abstracting this connectivity motif to GABA-ergic neuron inhibit a GABA-ergic pre-motor neuron instead of A31c inhibit A26f and add that the pre-pre-motor GAB-ergic neuron receives descending input.

We thank the reviewer for this suggestion. We revised the manuscript as suggested. Please see the response to Essential revision note described above.